# Physics informed neural networks for elliptic equations with oscillatory differential operators

**Arnav Gangal**                                                      *agangal00@gmail.com*
*Department of Mathematics*
*University of California, Los Angeles*

**Luis Kim**                                                          *luisyookim02@gmail.com*
*Department of Mathematics*
*The University of Texas at Austin*

**Sean P. Carney**                                                    *spcarney@math.ucla.edu*
*Department of Mathematics*
*University of California, Los Angeles*

**Reviewed on OpenReview:** *https://openreview.net/forum?id=QfyVqvpg7u*

## Abstract

Physics informed neural network (PINN) based solution methods for differential equations have recently shown success in a variety of scientific computing applications. Several authors have reported difficulties, however, when using PINNs to solve equations with multiscale features. The objective of the present work is to illustrate and explain the difficulty of using standard PINNs for the particular case of divergence-form elliptic partial differential equations (PDEs) with oscillatory coefficients present in the differential operator. We show that if the coefficient in the elliptic operator $a^\epsilon(x)$ is of the form $a(x/\epsilon)$ for a 1-periodic coercive function $a(\cdot)$, then the Frobenius norm of the neural tangent kernel (NTK) matrix associated to the loss function grows as $1/\epsilon^2$. This implies that as the separation of scales in the problem increases, training the neural network with gradient descent based methods to achieve an accurate approximation of the solution to the PDE becomes increasingly difficult. Numerical examples illustrate the stiffness of the optimization problem.

## 1 Introduction

Recent developments in deep learning have shown great promise for advancing computational and applied mathematics (E, 2021). Physics informed neural networks (PINNs) have recently emerged as a popular method for scientific computation. Building off earlier work Lagaris et al. (1998); Psichogios & Ungar (1992), PINNs were introduced in Raissi et al. (2019) and seek to approximate the true solution of a differential equation by a neural network $u(x; \theta)$ parameterized by weights and biases $\theta$. Consider the following general partial differential equation (PDE) defined on some $\Omega \subset \mathbb{R}^n$ by a differential operator $\mathcal{N}$

$$\mathcal{N}[u](x) = f(x), \qquad x \in \Omega$$
$$u(x) = g(x), \qquad x \in \partial\Omega, \tag{1}$$

where $x$ and $\Omega$ represent some general space-time coordinate and domain, respectively. Here and throughout this paper, the boundary conditions are taken to be of Dirichlet type. By the uniform approximation theorem (Hornik et al., 1989; Leshno et al., 1993), given sufficient data, a neural network can uniformly approximate classical smooth solutions to (1) whenever they exist.

In practice, the network parameters $\theta$ are determined by minimizing a loss function $\mathcal{L}$ that enforces (1) to hold for a set of $N_c$ collocation and $N_b$ boundary points, $\{x_i\}_{i=1}^{N_c} \in \Omega$ and $\{s_i\}_{i=1}^{N_b} \in \partial\Omega$, respectively. The

loss is

$$\mathcal{L}(\theta) = \frac{1}{N_c} \sum_{i=1}^{N_c} \frac{1}{2} \Big| \mathcal{N}[u](x_i;\theta) - f(x_i) \Big|^2 + \frac{\lambda}{N_b} \sum_{i=1}^{N_b} \frac{1}{2} \Big| u(s_i;\theta) - g(s_i) \Big|^2, \tag{2}$$

where $\lambda$ is a tunable parameter that weighs the relative importance of the boundary conditions. We refer to Karniadakis et al. (2021) for a review of this methodology applied in a wide range of contexts in scientific computing.

Despite the success of PINNs in a wide variety of applications, numerous authors have reported difficulties applying the technique to problems with multiscale features. Some examples include a scalar, nonlinear hyperbolic equation from a model of two-phase immiscible fluid flow in porous media (Fuks & Tchelepi, 2020), which can support shock waves, and systems of ordinary differential equations governing chemical kinetics (Ji et al., 2021), which exhibit stiff dynamics that evolve over a wide range of time scales. Difficulties have also been reported for the Helmholtz equation (Wang et al., 2021a), as well as linear hyperbolic problems, for example for example the one-dimensional advection equation (Krishnapriyan et al., 2021) and the wave equation (Wang et al., 2021b).

The focus of the current work is to illustrate and explain the difficulty of using standard PINN solution methods for linear elliptic boundary value problems (BVPs) of the form

$$-\nabla \cdot \big( a^\epsilon(x) \nabla u^\epsilon(x) \big) = f(x), \qquad x \in \Omega$$
$$u^\epsilon(x) = g(x), \qquad x \in \partial\Omega, \tag{3}$$

where the coefficient tensor $a^\epsilon : \mathbb{R}^n \to \mathbb{R}^n$ is uniformly bounded and coercive in $\epsilon$ and assumed to consist of entries that contain frequencies on the order of $\epsilon^{-1}$ for $0 < \epsilon \ll 1$. The measure of the domain $\Omega \subset \mathbb{R}^n$ is assumed to be $\mathcal{O}(1)$, and hence, (3) is a multiscale problem that models, for example, steady-state heat conduction in a composite material or porous media flow governed by Darcy's law.

Theoretical analysis and numerical experiments presented below illustrate that, whenever standard PINN architectures are used in conjunction with gradient descent based training for the multiscale problem (3), the resulting optimization problem becomes increasingly difficult as the scale separation in the BVP increases, i.e. as $\epsilon$ vanishes. After motivating the present study in Section 2, we show in Section 3 that the neural tangent kernel matrix associated with the PINN approximation to (3) has a Frobenius norm that becomes unbounded as $\epsilon \downarrow 0$. Numerical examples in Section 4 illustrate that during training, the ordinary differential equation that governs the evolution of the BVP residuals indeed becomes increasingly stiff as $\epsilon \downarrow 0$, translating to poor PINN performance for problems with a large separation of scales.

## 2 Motivation

The motivation for attempting to use physics informed neural network solutions to the oscillatory problem (3) is to investigate whether a connection can be established between asymptotic homogenization theory and the so-called "frequency principle" in deep learning.

Recall from mathematical homogenization theory (Bensoussan et al., 2011) that, under suitable conditions, the solution to (3) is well approximated as $\epsilon \downarrow 0$ by the solution to a homogenized equation

$$-\nabla \cdot \big( \overline{a}(x) \nabla \overline{u}(x) \big) = f(x), \qquad x \in \Omega$$
$$\overline{u}(x) = g(x), \qquad x \in \partial\Omega.$$

Both the homogenized coefficients $\overline{a}$ and the solution $\overline{u}$ do not contain $\epsilon$-scale oscillations; the latter approximates the large-scale, low-frequency features of the oscillatory function $u^\epsilon$.

Additionally, when neural networks learn a target function, they are known to learn the low frequency components more rapidly than the large frequencies (Rahaman et al., 2019; Xu et al., 2020). This "frequency principle" was shown to hold for gradient descent training in Luo et al. (2021); Markidis (2021); empirically the result can be observed for PINNs applied to relatively simple problems (Wang et al., 2021b), even when

more widely used optimizers are used for training, e.g. Adam (Kingma & Ba, 2014). Consider as a brief representative example the PINN solution to the one-dimensional Poisson BVP

$$-\frac{d^2}{dx^2}u(x) = \sin(x) + \sin(5x) + \sin(15x) + \sin(55x) =: f(x) \tag{4}$$

for $x \in (-\pi, \pi)$ with homogeneous Dirichlet boundary conditions $u(-\pi) = u(\pi) = 0$. For collocation points $\{x_i\}_{i=1}^{N_c} \in (-\pi, \pi)$ the loss function becomes

$$\mathcal{L}(\theta) = \frac{1}{N_c}\sum_{i=1}^{N_c}\frac{1}{2}\left|\frac{d^2}{dx^2}u(x_i;\theta) + f(x_i)\right|^2 + \frac{1}{4}\lambda\left(\left|u(-\pi;\theta)\right|^2 + \left|u(\pi;\theta)\right|^2\right). \tag{5}$$

Figure 1 shows the evolution (as a function of training iteration) of the complex modulus of

$$\widehat{e}_k = (\widehat{u}_{\text{true}})_k - (\widehat{u}_{\text{NN}})_k \tag{6}$$

where $u_{\text{true}}$ is the true solution to the BVP, $u_{\text{NN}}$ is the neural network approximation, $\widehat{\cdot}$ denotes the discrete Fourier transform, and the index $k$ denotes the frequency of the coefficient. A rolling average is used to make the trajectories more legible. The PINN solution is computed with a fully connected neural network with four hidden layers of sixty nodes each, $N_c = 512$ equispaced collocation points, and $\lambda = 80$; see Appendix A for a complete description of the training process that generates the final neural network output. One can clearly observe the frequency principle in this simple example; the low frequencies of the target BVP solution are more rapidly learned than the high frequencies. See also Markidis (2021) for another example in a two-dimensional Poisson problem, as well as Wang et al. (2021b) for mathematical analysis of this phenomenon for Poisson equations.

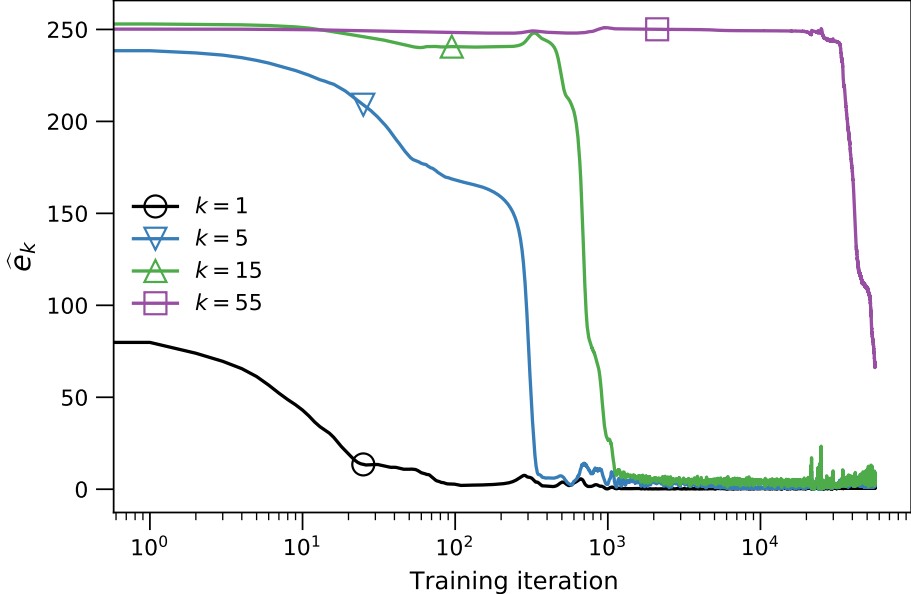

Figure 1: A simple illustration of the frequency principle: evolution as a function of training iteration of the magnitude of the discrete Fourier transform of the error (6) between the true solution to the BVP (4) and the neural network approximation.

Given some evidence that PINNs can learn low frequency features of PDE solutions, it is reasonable to ask if the homogenized solution 'naturally' arises when trying to learn solutions to the multiscale problem (3), and, if not, for what reason? The homogenized solution itself is of course useful in a variety of applications; it additionally could be used to construct coarse grid solutions for a multigrid solver of the full, oscillatory

problem as in Engquist & Luo (1997). The natural smoothing properties of standard iterative solvers, e.g. the method of Gauss-Seidel or successive over-relaxation (Briggs et al., 2000), would then rapidly reduce the error of high-frequency solution components that are not easily approximated by the neural networks; such a 'hybrid' multigrid strategy was explored in Markidis (2021) for Poisson equations.

Recent theoretical results in Shin et al. (2020) guarantee that as the number of collocation $N_c$ and boundary $N_b$ points tend to infinity, the sequence of minimizers to a "Hölder regularized" version of the loss function (2) will converge to classical solutions of elliptic and parabolic PDEs (whenever they exist) in the limit of infinite learning data. Nevertheless, the authors in both Han & Lee (2023) and Leung et al. (2022) reported difficulties training neural networks to achieve such minimizers of the loss function associated to the multiscale problem (3). The purpose of the current work is to characterize the reason why optimization is difficult with both theory and numerical examples.

The theoretical results presented below describe the neural tangent kernel matrix of the PINN solution to the BVP (3); this matrix appears in the ordinary differential equation that, under gradient descent training dynamics, governs the evolution of the PDE and boundary residuals that define the PINN loss function. Under a few technical assumptions, we show that the matrix Frobenius norm becomes unbounded as $\epsilon \downarrow 0$. This result, along with numerical evidence presented in Section 4, suggests that as that as the scale separation in the differential operator in (3) increases, the optimization problem that determines the neural network approximation to the PDE solution becomes increasingly stiff.

## 3 Neural tangent kernel matrix theory

### 3.1 Theoretical results

We now describe the neural tangent kernel (NTK) matrix for the physics informed neural network approximation to the multiscale elliptic equation (3) before showing that it can grow arbitrarily large in norm as $\epsilon \downarrow 0$. See Jacot et al. (2018) for the original development of the NTK theory for least-squares regression problems, as well as Wang et al. (2021b) and Wang et al. (2022) for an extension of the theory to PINNs.

For the presentation below it will be useful to define the linear multiscale differential operator

$$L^\epsilon \varphi := -\nabla \cdot \left( a^\epsilon \nabla \varphi \right),$$

where $\varphi$ is some suitably regular function. The multiscale elliptic boundary value problem (3) then becomes

$$
\begin{aligned}
L^\epsilon u^\epsilon &= f & \Omega \\
u^\epsilon &= g & \partial\Omega.
\end{aligned}
\tag{7}
$$

For simplicity, we assume throughout that the entries of the coefficient tensor $a^\epsilon$ are at least once continuously differentiable, and hence bounded on any compact domain $\Omega$.

Given some neural network $u(x; \theta)$ parameterized by $N_p$ weights and biases $\theta$, as well as collocation $\{x_i\}_{i=1}^{N_c} \in \Omega$ and boundary points $\{s_i\}_{i=1}^{N_b} \in \partial\Omega$, the loss function associated to (7) becomes

$$
\mathcal{L}^\epsilon(\theta) := \frac{1}{N_c} \sum_{i=1}^{N_c} \frac{1}{2} \left| L^\epsilon u(x_i; \theta) - f(x_i) \right|^2 + \frac{\lambda}{N_b} \sum_{i=1}^{N_b} \frac{1}{2} \left| u(s_i; \theta) - g(s_i) \right|^2.
\tag{8}
$$

If the residual values

$$
r_{\mathrm{pde}}(x_i; \theta) := L^\epsilon u(x_i; \theta) - f(x_i), \qquad i = 1, \ldots, N_c
\tag{9}
$$

and

$$
r_{\mathrm{b}}(s_i; \theta) := u(s_i; \theta) - g(s_i), \qquad i = 1, \ldots, N_b
\tag{10}
$$

are grouped together into a single vector $y(\theta)$, and if the parameters $\theta$ evolve according to the gradient flow

$$
\frac{d\theta}{dt} = -\nabla_\theta \mathcal{L}^\epsilon,
\tag{11}
$$

then $y(\theta(t))$ will evolve according to an initial value problem of the form

$$\frac{d}{dt}y(t) = -K^\epsilon(y(t))\,y(t),\tag{12}$$

where $K^\epsilon$ is called the neural tangent kernel matrix. Note here that both $\theta$ and $y$ implicitly depend on $\epsilon$; however, the dependence is not explicitly marked.

We next derive (12), noting again that similar results can be found in Wang et al. (2021b) and Wang et al. (2022).

**Lemma 3.1.** *For neural network parameters $\theta \in \mathbb{R}^{N_p}$, let $y(\theta) \in \mathbb{R}^{N_c+N_b}$ be the vector of residual values*

$$y(\theta) = \big(r_{\mathrm{pde}}(x_1;\theta),\ldots,r_{\mathrm{pde}}(x_{N_c};\theta), r_{\mathrm{b}}(s_1;\theta),\ldots,r_{\mathrm{b}}(s_{N_b};\theta)\big)^T$$

*where the entries $r_{\mathrm{pde}}$ and $r_{\mathrm{b}}$ are defined by (9) and (10). Suppose that the parameters $\theta$ evolve from some initial value $\theta_0$ according to the gradient flow (11). Then $y(\theta(t))$ evolves from the initial condition $y(\theta_0)$ according to*

$$\frac{d}{dt}y(t) = -K^\epsilon(y(t))\,y(t),\tag{13}$$

*where the explicit dependence on $\theta$ in (13) is dropped for convenience. The neural tangent kernel matrix is given by*

$$K^\epsilon(t) = \begin{pmatrix} K_{uu}^\epsilon(t) & K_{ub}^\epsilon(t) \\ K_{bu}^\epsilon(t) & K_{bb}^\epsilon(t) \end{pmatrix} \in \mathbb{R}^{(N_c+N_b)\times(N_c+N_b)},\tag{14}$$

*where the subblocks have entries*

$$(K_{uu}^\epsilon)_{ij}(t) = \frac{1}{N_c}\sum_{l=1}^{N_p} \frac{\partial}{\partial\theta_l}L^\epsilon u(x_i;\theta(t))\frac{\partial}{\partial\theta_l}L^\epsilon u(x_j;\theta(t)), \qquad 1 \le i,j \le N_c,$$

$$(K_{bb})_{ij}(t) = \frac{\lambda}{N_b}\sum_{l=1}^{N_p} \frac{\partial}{\partial\theta_l}u(s_i;\theta(t))\frac{\partial}{\partial\theta_l}u(s_j;\theta(t)), \qquad 1 \le i,j \le N_b,$$

*and*

$$(K_{ub}^\epsilon)_{ij}(t) = \frac{\lambda}{N_b}\sum_{l=1}^{N_p} \frac{\partial}{\partial\theta_l}L^\epsilon u(x_i;\theta(t))\frac{\partial}{\partial\theta_l}u(s_j;\theta(t)), \qquad 1 \le i \le N_c,\ 1 \le j \le N_b,$$

*while $K_{bu}^\epsilon$ is simply the scaled transpose of $K_{ub}^\epsilon$:*

$$(K_{bu}^\epsilon)_{ij} = \frac{N_b}{\lambda}\frac{1}{N_c}\,(K_{ub}^\epsilon)_{ji}, \qquad 1 \le i \le N_c,\ 1 \le j \le N_b.$$

The proof is a computation that follows from the chain rule from differential calculus; it can be found in Appendix B. We note that while the $K_{uu}^\epsilon$ and $K_{bb}^\epsilon$ subblocks of the NTK matrix are symmetric, overall, $K^\epsilon$ is not symmetric unless $N_b = \lambda N_c$.

Next we show that the NTK matrix (14) associated to the multiscale elliptic problem–in particular the $K_{uu}^\epsilon$, $K_{ub}^\epsilon$ and $K_{bu}^\epsilon$ subblocks–can become arbitrarily large for vanishing $\epsilon$. In contrast, because the boundary condition $g$ in (7) is indepedent of $\epsilon$, there is no explicit $\epsilon$-dependence in the $K_{bb}$ subblock.

For simplicity we assume the problem is one-dimensional ($n = 1$) and that the multiscale coefficient $a^\epsilon(x) = a(x/\epsilon)$ is $\epsilon$-periodic. After showing the desired result for $n = 1$, we briefly sketch below an additional proof for the higher dimension case $n > 1$ under additional technical assumptions.

In dimension $n = 1$, (7) becomes

$$-\frac{d}{dx}\left(a(x/\epsilon)\frac{d}{dx}u^\epsilon(x)\right) = f(x), \qquad x \in (a,b)$$

$$u^\epsilon(a) = u_a, \qquad u^\epsilon(b) = u_b.\tag{15}$$

The loss function associated to (15) is equation (8) with $N_b = 2$ and elliptic operator $L^\epsilon$ given by

$$L^\epsilon u(x;\theta) = -\left(a(x/\epsilon)\frac{d^2}{dx^2}u(x;\theta) + \frac{1}{\epsilon}a'(x/\epsilon)\frac{d}{dx}u(x;\theta)\right), \tag{16}$$

where it is assumed the oscillatory function $a$ is differentiable. The $1/\epsilon$ factor in the second term will be the source of the divergent behavior of the NTK as $\epsilon \downarrow 0$.

We first show the result for a neural network with one hidden layer before extending it to more general fully connected networks.

**Theorem 3.2.** *Let $u : \mathbb{R} \to \mathbb{R}$ be a neural network with one hidden layer of width $d$ and smooth activation function $\sigma$, so that*

$$u(x;\theta) = \sum_{k=1}^{d} W_k^{(1)}\sigma\left(W_k^{(0)}x + b_k^{(0)}\right) + b^{(1)}, \tag{17}$$

*where $b^{(1)} \in \mathbb{R}$, $W^{(1)} \in \mathbb{R}^{1 \times d}$, and $W^{(0)}, b^{(0)} \in \mathbb{R}^{d \times 1}$, and assume that $\forall x \in \mathbb{R}$*

$$0 < \sigma'(x) < \infty. \tag{18}$$

*Let $a(y)$ be a one-periodic, non-constant $C^1$ function that is bounded and coercive, so that $\forall y \in \mathbb{R}$, there exist some $\lambda, \Lambda \in \mathbb{R}$ such that*

$$0 < \lambda \leq a(y) \leq \Lambda,$$

*and let $a^\epsilon(x) = a(x/\epsilon)$. For a fixed number of collocation points $N_c$, let $K^\epsilon$ be the neural tangent kernel matrix (14) associated to the loss function (8) of the boundary value problem (15), and suppose the network parameters $\theta$ evolve according to the gradient flow (11) for $0 \leq t \leq T$ for some $T > 0$.*

*Suppose also that the parameters $\theta$ evolving according to (11) are bounded uniformly in both $t$ and $\epsilon$:*

$$\sup_{0 < \epsilon < \epsilon_0} \sup_{0 \leq t \leq T} \|\theta(t)\|_\infty < C \tag{19}$$

*for some $\epsilon_0 > 0$. Finally, let $\Upsilon$ be the set of $t \in [0, T]$ for which there exists at least one entry of $W^{(0)}$ that is asymptotically larger than $\epsilon$; that is, $t \in \Upsilon$ if and only if*

$$\lim_{\epsilon \downarrow 0} \epsilon/W_l^{(0)}(t) = 0$$

*for some $1 \leq l \leq d$. Then $\forall t \in \Upsilon$*

$$\lim_{\epsilon \downarrow 0} \|K^\epsilon(t)\|_F = \infty$$

*where $\|\cdot\|_F$ is the Frobenius norm.*

Before proving the result, we briefly remark on the theorem assumptions. First note that the number of collocation points $N_c$ is assumed to be fixed. Since we will show that the entries of the NTK matrix are proportional to

$$\frac{1}{N_c}\frac{1}{\epsilon^2},$$

the singular nature of the evolution of the residuals (12) could in principle be prevented by taking $N_c \sim \epsilon^{-2}$. However, this can make the computational cost of gradient descent based optimization prohibitively expensive whenever the problem scale separation is sufficiently large.

Next, note the assumption (18) on $d\sigma/dx$ holds for sigmoidal-type activation functions commonly used for PINNs (Lu et al., 2021b), such as the hyberbolic tangent and logistic functions. Although it does not hold for the ReLU function $\sigma(x) = \max(0, x)$, these are of course not suitable for PINNs and other NN based solution methods that are based on the strong formulation of PDEs with second order (or higher) differential operators.

Regarding the set $\Upsilon \subseteq [0, T]$, note that if $t \notin \Upsilon$, then each entry of $W^{(0)}$ will converge to 0 in the limit $\epsilon \downarrow 0$; in this scenario the neural network (17) would limit to a constant function and hence not be a suitable

solution to (15) in general. Finally, if the uniform bound (19) on the network parameters $\theta$ did not hold, then an unstable, divergent training process could result (Wang et al., 2022); the bound helps prevent the so-called "vanishing gradient" problem that can occur when components of $\nabla_\theta \mathcal{L}^\epsilon$ vanish, preventing descent in that direction.

*Proof of Theorem 3.2.* The proof, like that of Lemma 3.1, is a consequence of direct computation; note that it is sufficient for a single entry of a matrix to diverge to ensure the Frobenius norm diverges. In particular we show that entries in the $K_{uu}^\epsilon$ subblock of the full NTK matrix $K^\epsilon$ scale as $1/(N_c \, \epsilon^2)$.

Indeed, recall from Lemma 3.1 that

$$(K_{uu}^\epsilon)_{\alpha\alpha}(t) = \frac{1}{N_c} \sum_{\gamma=1}^{N_p} \left( \frac{\partial}{\partial\theta_\gamma} L^\epsilon u(x_\alpha; \theta(t)) \right)^2 \tag{20}$$

for any $1 \leq \alpha \leq N_c$. Since (20) is a sum of squares, if just one entry in the sum blows up as $\epsilon \downarrow 0$, then the entire sum of course will as well. Below we drop the explicit dependence of the parameters $\theta$ on $t$ for notational convenience.

First, let $g(x; \theta)$ be the $\mathbb{R}^d$ valued function whose $k$-th component equals

$$g_k(x; \theta) = \sigma\left( W_k^{(0)} x + b_k^{(0)} \right),$$

so that

$$u(x; \theta) = \sum_{k=1}^d W_k^{(1)} g_k(x; \theta)) + b^{(1)}.$$

By linearity,

$$\frac{\partial}{\partial W_\gamma^{(1)}} L^\epsilon u(x; \theta) = L^\epsilon g_\gamma(x; \theta)$$

for any $1 \leq \gamma \leq d$. So, to prove the desired result, it suffices to show that

$$\frac{1}{N_c} \left[ L^\epsilon g_\gamma(x_\alpha; \theta) \right]^2 \to \infty \tag{21}$$

as $\epsilon \downarrow 0$. Using (16), the left-hand side of (21) equals

$$\frac{1}{N_c} \left[ \left( a(x_\alpha/\epsilon) \frac{d^2}{dx^2} g_\gamma(x_\alpha; \theta) \right)^2 + \frac{2}{\epsilon} \left( a(x_\alpha/\epsilon) \frac{d^2}{dx^2} g_\gamma(x_\alpha; \theta) \, a'(x_\alpha/\epsilon) \frac{d}{dx} g_\gamma(x_\alpha; \theta) \right) \right.$$
$$\left. + \frac{1}{\epsilon^2} \left( a'(x_\alpha/\epsilon) \frac{d}{dx} g_\gamma(x_\alpha; \theta) \right)^2 \right]. \tag{22}$$

Since $a$ is a periodic $C^1$ function, both $a(x_\alpha/\epsilon)$ and $a'(x_\alpha/\epsilon)$ are bounded independent of $x_\alpha/\epsilon \in \mathbb{R}$. The derivatives of $g_\gamma$ are

$$\frac{d}{dx} g_\gamma(x; \theta) = \sigma'\left( W_\gamma^{(0)} x + b_\gamma^{(0)} \right) W_\gamma^{(0)} \tag{23}$$

and

$$\frac{d^2}{dx^2} g_\gamma(x; \theta) = \sigma''\left( W_\gamma^{(0)} x + b_\gamma^{(0)} \right) \left( W_\gamma^{(0)} \right)^2. \tag{24}$$

Since the activation function is smooth and the network parameters are uniformly bounded in $\epsilon$ and $t$, both (23) and (24) are also bounded for all $x_\alpha \in [a, b]$. Consequently, (22) is dominated by the $1/\epsilon^2$ term for $\epsilon$ vanishingly small. The positivity assumption (18) on $d\sigma/dx$ and the uniform boundedness of the network parameters imply also that the (absolute value of the) first derivative (23) is bounded below by a constant independent of $\epsilon$; this implies that for any $t \in \Upsilon$

$$\lim_{\epsilon \downarrow 0} \frac{1}{\epsilon} \left| \frac{d}{dx} g_l(x; \theta) \right| = \infty$$

holds for at least one $l \in \{1, \ldots d\}$. Take $\gamma = l$, and let $\epsilon$ vanish monotonically to zero in such a way that $a'(x_\alpha/\epsilon) \neq 0$ for any $\epsilon$ (such a sequence exists since $a$ is periodic and non-constant). Then (22) indeed limits to positive infinity, giving the desired result.

$\square$

Without giving a formal proof, we now briefly describe sufficient conditions under which Theorem 3.2 could be extended to the more general case $n > 1$ (assuming one retains the positivity assumption (18) on $d\sigma/dx$ and the uniform boundedness assumption (19) on the network parameters). A neural network with one hidden layer of width $d$ would be

$$u(x; \theta) = \sum_{k=1}^{d} W_k^{(1)} \sigma\Big(\sum_{l=1}^{n} W_{kl}^{(0)} x_l + b^{(0)}\Big) + b^{(1)};$$

here $x_l$ denotes the $l$-th component of the function input $x \in \mathbb{R}^n$. Following the same argument just given, define

$$g_k(x; \theta) = \sigma\Big(\sum_{l=1}^{n} W_{kl}^{(0)} x_l + b^{(0)}\Big),$$

and assume that there exists some $1 \leq \mu \leq d$ and $1 \leq \nu \leq n$ such that $W_{\mu\nu}^{(0)}$ is asymptotically larger than $\epsilon$ (as before, if no such entries exist, then $W^{(0)}$ limits to the zero matrix as $\epsilon \downarrow 0$, and hence $u(x; \theta)$ limits to a constant function). The Frobenius norm of the PINN's neural tangent kernel matrix will blow-up if $[L^\epsilon g_\mu(x; \theta)]^2$ blows-up at some collocation point $x$. This can occur if the entries for which $\nabla \cdot a$ are nonzero coincide with the entries where $W^{(0)}$ is larger than $\mathcal{O}(\epsilon)$; more precisely, if there is a sequence $\epsilon \downarrow 0$ such that

$$\sum_{i=1}^{n} \frac{\partial a_{i\nu}}{\partial x_i}\Big(\frac{x}{\epsilon}\Big) \sigma'\Big(\sum_{l=1}^{n} W_{\mu l}^{(0)} x_l + b_\mu^{(0)}\Big) W_{\mu\nu}^{(0)} \neq 0$$

for any collocation point $x$, then the main result will generalize.

Returning to the case of one-dimension ($n = 1$), we now extend Theorem 3.2 to more general fully connected neural networks under an additional assumption on the network parameter's asymptotic behavior.

**Theorem 3.3.** *Let $u : \mathbb{R} \to \mathbb{R}$ be a fully connected neural network with $\Lambda$ hidden layers of widths $d_1, d_2, \ldots, d_\Lambda$, so that*

$$
\begin{aligned}
u^{(0)}(x) &= x, \\
u^{(l)}(x) &= \sigma\big(W^{(l)} u^{(l-1)}(x) + b^{(l)}\big), \qquad 1 \leq l \leq \Lambda, \\
u(x; \theta) &= W^{(\Lambda+1)} \cdot u^{(\Lambda)}(x) + b^{(\Lambda+1)}.
\end{aligned}
\tag{25}
$$

*If all of the assumptions from Theorem 3.2 are retained, but $\Upsilon$ is now defined to be the set of $t \in [0, T]$ such that the magnitude of every entry of each matrix $W^{(l)}$, $1 \leq l \leq \Lambda$, are asymptotically larger than $\epsilon^{1/\Lambda}$, then $\forall t \in \Upsilon$*

$$\lim_{\epsilon \downarrow 0} \|K^\epsilon(t)\|_F = \infty,$$

*where $K^\epsilon$ is the NTK matrix associated to the loss function* (8) *of the boundary value problem* (15).

The proof proceeds nearly identically to that of Theorem 3.2, and hence it is in Appendix C.

An immediate corollary of Theorem 3.3 is that the spectral radius of $K_{uu}^\epsilon$ must also blow up as $\epsilon \downarrow 0$; in contrast, because there is no explicit $\epsilon$-dependence in the $K_{bb}$ subblock, one expects its spectral radius to be bounded independently of $\epsilon$, which is indeed numerically observed in section 4.1.

**Corollary 3.4.** *Let $\rho(K_{uu}^\epsilon(t))$ denote the spectral radius of $K_{uu}^\epsilon(t)$. Under the same assumptions as Theorem 3.3, we have for any $t \in \Upsilon$ that*

$$\lim_{\epsilon \downarrow 0} |\rho(K_{uu}^\epsilon(t))| = \infty.$$

See Appendix D for the simple proof.

### 3.2 Discussion

Fundamentally, using PINNs to numerically solve PDEs involves optimizing a multiobjective loss functional, for example of the form

$$\mathcal{L}(\theta) = \mathcal{L}_{\text{PDE}}(\theta) + \mathcal{L}_{\text{BC}}(\theta)$$

for boundary values problems. The present work builds on the previous studies Wang et al. (2021a), Wang et al. (2021b) and Wang et al. (2022), where certain PINN failure modes were identified for multiscale problems that, broadly speaking, involve inbalances between $\mathcal{L}_{\text{PDE}}$ and $\mathcal{L}_{\text{BC}}$.

These previous analyses focused on unsatisfactory PINN training in the particular case of Poisson-type PDEs, i.e. where the differential operator is the Laplacian (or, in the one-dimensional case, the second derivative). The multiscale character of the problems originated from oscillatory forcing functions; an illustrative example is the boundary value problem

$$- \frac{d^2}{dx^2} u^\epsilon(x) = \sin(x) + (1/\epsilon)\sin(x/\epsilon) \tag{26}$$

on $[0,1]$ along with appropriate boundary conditions (see e.g. Eq. (2.6) in Wang et al. (2021b)). The present work focuses on elliptic problems that feature a more general class of divergence-form differential operators with oscillatory coefficients $a^\epsilon(x)$, as in Eq. (3). The differences between the two cases can be considerable.

For example, from Wang et al. (2021a) it is known that PINNs can fail to train when the magnitudes of the gradients of the two different loss components ($\mathcal{L}_{\text{PDE}}$ and $\mathcal{L}_{\text{BC}}$) with respect to the network parameters $\theta$ are imbalanced, i.e. when either $|\nabla_\theta \mathcal{L}_{\text{PDE}}| \gg |\nabla_\theta \mathcal{L}_{\text{BC}}|$ or $|\nabla_\theta \mathcal{L}_{\text{PDE}}| \ll |\nabla_\theta \mathcal{L}_{\text{BC}}|$. For multiscale Poisson equations such as (26), the $1/\epsilon$ term in the right-hand side forcing results in $|\nabla_\theta \mathcal{L}_{\text{PDE}}|$ scaling like $1/\epsilon$. In contrast, for Darcy-type problems such as (3) (or, in one-dimension, (15)), $|\nabla_\theta \mathcal{L}_{\text{PDE}}|$ is more singular; it scales like $1/\epsilon^2$. In both cases, $|\nabla_\theta \mathcal{L}_{\text{BC}}|$ is indepedent of $\epsilon$, so the discrepancy is greater for problems considered in the present work.

An alternative PINN failure mode analyzed in Wang et al. (2021b) and Wang et al. (2022) is when the the eigenvalues of the different NTK matrix subblocks differ greatly in modulus. The authors reported a large imbalance between the eigenvalues of the $K_{uu}$ and $K_{bb}$ subblocks for Poisson-style problems. However, for equations such as (26), it is clear from the definitions in (14) that the $K_{uu}$ subblock is independent of $\epsilon$, since the forcing function $f$ (which is responsible for the equation's multiscale character) is independent of $\theta$.

In contrast, the results from Section 3.1 show that while the $K_{bb}$ subblock is independent of $\epsilon$ for Darcy-type problems, the spectral radius of the $K_{uu}$ subblock increases as $1/\epsilon^2$. Hence, for problems with large scale separation, the imbalance in eigenvalues is even more severe than in the Poisson case. For this reason, training PINNs with gradient based optimizers to achieve an accurate approximation to solutions to multiscale equations of the form (3) becomes increasingly difficult as a function of increasing problem scale separation.

A common strategy to deal with the imbalances just highlighted is to assign the different loss components $\mathcal{L}_{\text{PDE}}$ and $\mathcal{L}_{\text{BC}}$ different weights $\lambda_{\text{PDE}}$ and $\lambda_{\text{BC}}$; this was proposed e.g. in Wang et al. (2021a), van der Meer et al. (2022) and Wang et al. (2022). It is reasonable, for example, for the equations considered here to try and weight $\mathcal{L}_{\text{PDE}}$ by $\lambda_{\text{PDE}} = \epsilon^2$ to cancel the $1/\epsilon^2$ factor in the NTK $K_{uu}^\epsilon$ subblock. Although this of course changes the magnitude of the eigenvalues of the NTK subblock, it does not change their distribution, and hence cannot directly resolve spectral bias Wang et al. (2022), which may explain the lack of success using adaptive weighting strategies that was reported in both Leung et al. (2022) and Han & Lee (2023). The numerical examples in Section 4.4 below illustrate that even under this rescaling, the stiffness of the gradient flow dynamics described by (11) remains, translating to poor PINN performance.

## 4 Numerical results

### 4.1 Scaling of the neural tangent kernel matrix

Theorems 3.2 and 3.3 state that the Frobenius norm of the neural tangent kernel (NTK) matrix $K^\epsilon$ associated to the PINN approximation of the multiscale BVP (15) should scale inversely proportional to $N_c \epsilon^2$ for

gradient descent training with infinitesimal time steps. Here we numerically reproduce this scaling for a neural network trained in a more practical setting, namely with the Adam optimizer (Kingma & Ba, 2014).

As in Theorems 3.2 and 3.3, we focus on the $K_{uu}^\epsilon$ subblock of the NTK matrix. For a neural network with one hidden layer $d$, the $ij$-th entry of $K_{uu}^\epsilon$ is given by

$$\left(K_{uu}^\epsilon\right)_{ij}(t) = \sum_{k=1}^{d} \left( \frac{\partial}{\partial W_k^{(1)}} L^\epsilon u(x_i; \theta(t)) \frac{\partial}{\partial W_k^{(1)}} L^\epsilon u(x_j; \theta(t)) + \frac{\partial}{\partial W_k^{(0)}} L^\epsilon u(x_i; \theta(t)) \frac{\partial}{\partial W_k^{(0)}} L^\epsilon u(x_j; \theta(t)) \right.$$
$$\left. + \frac{\partial}{\partial b_k^{(0)}} L^\epsilon u(x_i; \theta(t)) \frac{\partial}{\partial b_k^{(0)}} L^\epsilon u(x_j; \theta(t)) \right)$$

(note $\partial L^\epsilon u(x; \theta)/\partial b^{(1)} = 0$). For network width $d = 50$ and $N_c = 256$ collocation points, Figure 2(a) shows the Frobenius norm $\|K_{uu}^\epsilon\|_F$ associated to (15) for the particular case that $[a, b] = [-\pi, \pi]$, $u_a = u_b = 0$, and $a^\epsilon(x) = 1/(2.1 + 2\sin(2\pi x/\epsilon))$. The norm is computed at time $t = 0$, i.e. at initialization, for a sequence of $\epsilon$ values between $1/10$ and $1/100$. For every $\epsilon$ value, the neural network parameters are initialized from the normal distribution with mean zero and unit variance. Figure 2(b) shows the Frobenius norms after initializing with the Glorot distribution (Glorot & Bengio, 2010) and then training for ten thousand iterations with the Adam optimizer at a learning rate of $\eta = 10^{-5}$. In both cases, the norm increases as $1/\epsilon^2$, consistent with the theory developed above.

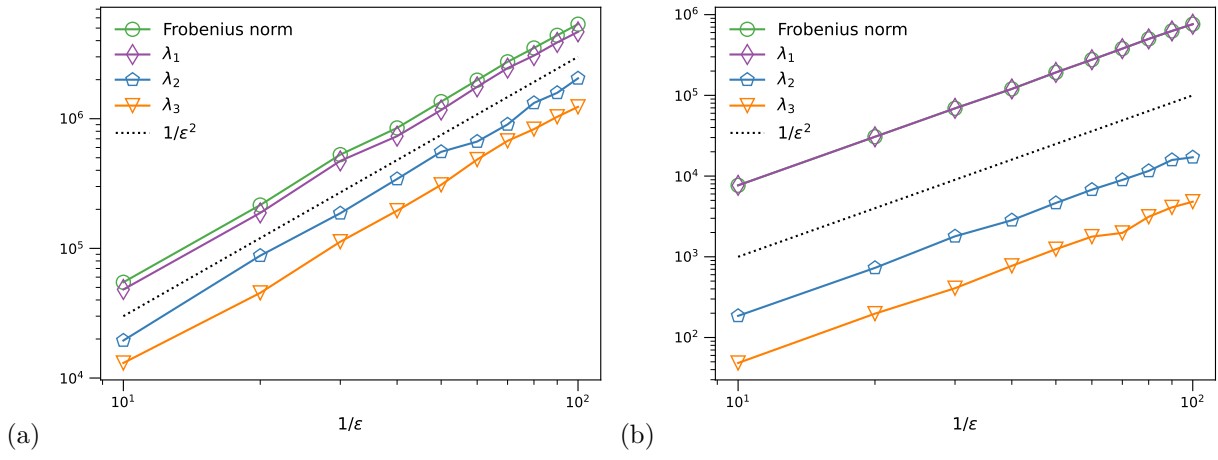

(a)                                                               (b)

Figure 2: Scaling of the Frobenius norm of the NTK matrix and the first three principal eigenvalues (a) at initialization and (b) after training with Adam optimizer.

Recall that the Frobenius norm of a square matrix is equal to the Euclidean norm of its singular values. Since $K_{uu}^\epsilon$ is a symmetric matrix, its singular values equal the absolute value of its (real valued) eigenvalues. Hence, Theorems 3.2 and 3.3 imply that $\sum_{i=1}^{N_c} (\lambda_i^\epsilon)^2 \to \infty$ as $\epsilon$ vanishes, where $\lambda_i^\epsilon$ are the eigenvalues of $K_{uu}^\epsilon$. Figures 2(a) and 2(b) further suggest that, for the BVP just described, the first few principal eigenvalues (denoted $\lambda_1, \lambda_2$, and $\lambda_3$) also increase at a rate proportional to $1/\epsilon^2$, both at initialization and after training with the Adam optimizer. In contrast, the spectral radius of the $K_{bb}$ subblock (not shown) is essentially independent of $\epsilon$.

For the particular case $\epsilon = 1/80$, Figure 3 shows the full spectra of $K_{uu}^\epsilon$ both at initialization and after training. At initialization, there are a handful of large eigenvalues, but the rest are clustered around the origin; after training, there is a single large eigenvalue well-separated from the cluster near the origin. In both cases the ratio $\lambda_{\max}^\epsilon/\lambda_{\min}^\epsilon$ is quite large, indicating that (12) is a stiff initial value problem.

### 4.2 Learning a two-scale function

Consider next the two-scale function

$$u^\epsilon(x) = \sin(2x) + \epsilon \sin(x/\epsilon) - \frac{\epsilon}{\pi} \sin(\pi/\epsilon)\, x. \tag{27}$$

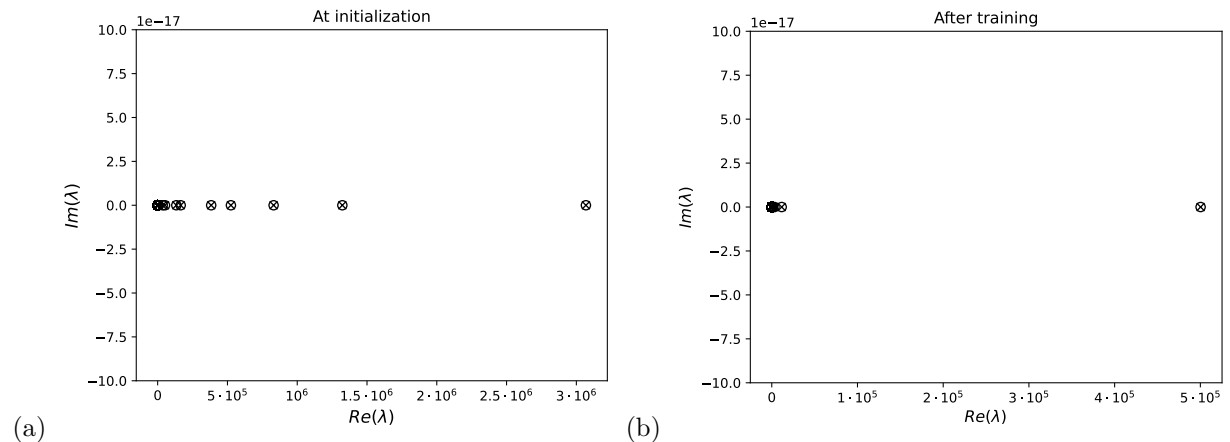

Figure 3: The eigenspectrum of $K_{uu}^\epsilon$ (a) at initialization and (b) after training.

This function satisfies the Poisson BVP

$$-\frac{d^2}{dx^2}u^\epsilon(x) = 4\sin(2x) + \frac{1}{\epsilon}\sin(x/\epsilon), \qquad -\pi < x < \pi \qquad (28)$$

with homogeneous Dirichlet boundary conditions $u^\epsilon(-\pi) = u^\epsilon(\pi) = 0$; it additionally satisfies the elliptic, Darcy BVP given by

$$-\frac{d}{dx}\Big(a(x/\epsilon)\frac{d}{dx}u^\epsilon(x)\Big) = f^\epsilon(x), \qquad (29)$$

for $x \in (-\pi, \pi)$, again with homogeneous Dirichlet conditions, where

$$a(x/\epsilon) = \big(2.1 + 2\sin(x/\epsilon)\big)^{-1}$$

and the expression for the right-hand side forcing $f^\epsilon(x)$ is listed in Appendix E.

We describe three different neural network approximations to the two-scale function $u^\epsilon(x)$ by:

(1) regression (i.e. supervised learning),

(2) a PINN solver for the Poisson problem (28), and

(3) a PINN solver for the Darcy problem (29).

We first report the results for each case when $\epsilon = 1/32$ and then repeat the experiments in Section 4.3 at smaller values of $\epsilon$ to observe the behavior of each approximation as a function of increasing scale separation.

For $\epsilon = 1/32$, the first two methods listed above generate satisfactory approximations to (27), indicating that (a good approximation to) the target function lives in the "span" of the neural network used; the third method does not. All of the numerical tests are implemented in the open-source DeepXDE package (Lu et al., 2021b), and in each case, the neural network weights and biases are randomly initialized with the Glorot distribution (Glorot & Bengio, 2010).

Consider first approximating $u^\epsilon(x)$ with a neural network by regression, which we denote as $u_R(x; \theta)$. For $\{x_i\}_{i=1}^N \in [-\pi, \pi]$, the network parameters are determined by minimizing

$$\mathcal{L}^\epsilon(\theta) = \frac{1}{N}\sum_{i=1}^N \big(u_{\mathrm{R}}(x_i; \theta) - u^\epsilon(x_i)\big)^2. \qquad (30)$$

We set $\epsilon = 1/32$ and use a neural network with hyperbolic tangent activation functions and four hidden layers, each of width $d = 40$. Using $N = \lceil 2\pi/(\epsilon/2)\rceil = 403$ equispaced training points, the network is trained

with the Adam (Kingma & Ba, 2014) and L-BFGS (Byrd et al., 1995) optimizers for ten independent trials. In each test, Adam is used for $20,000$ and $10,000$ iterations with learning rates of $\eta = 10^{-2}$ and $\eta = 10^{-3}$, respectively, and L-BFGS is then used, on average, for approximately 3600 steps.

Let $\mathbb{E}\, u_\mathrm{R}(x;\theta)$ denote the neural network averaged over all ten trials. Taking $\{x_i\}_{i=1}^{1000}$ to be a set of equispaced points from $-\pi$ to $\pi$, Figures 4(a) and (b) show the resulting approximation and error, respectively. The maximum error is approximately

$$\max_{x_i} |e_\mathrm{R}| := \max_{x_i} \left| u^\epsilon(x_i) - \mathbb{E}\, u_\mathrm{R}(x_i;\theta) \right| \approx 2.65 \cdot 10^{-3},$$

and the average variance in the error over the interval $[-\pi, \pi]$ is about $3 \cdot 10^{-7}$.

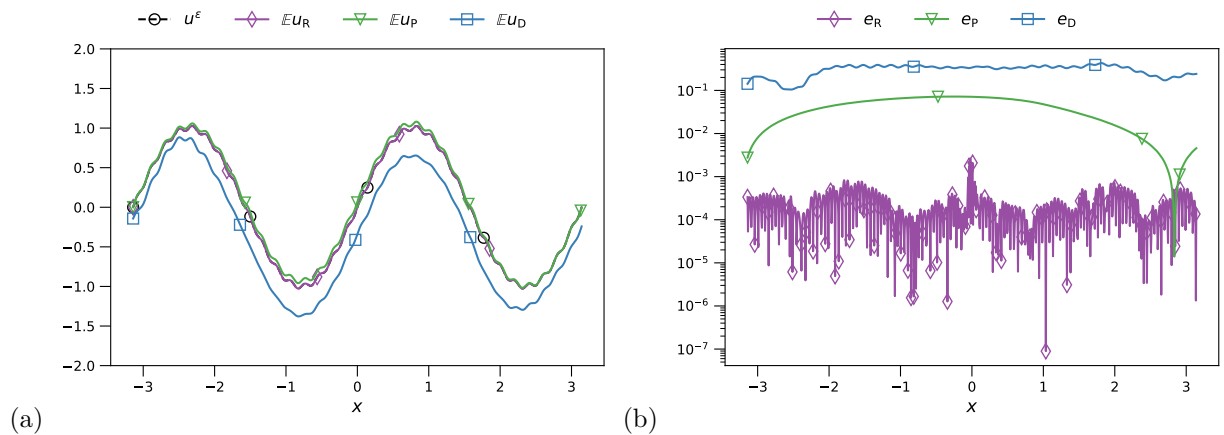

(a)
(b)

Figure 4: (a) For $\epsilon = 1/32$, the two-scale function $u^\epsilon(x)$ given by (27) and its approximations by regression ($\mathbb{E}\, u_\mathrm{R}$), a Poisson PINN ($\mathbb{E}\, u_\mathrm{P}$) and a Darcy PINN ($\mathbb{E}\, u_\mathrm{D}$); in each case the neural network contains four hidden layers of width $d = 40$. (b) The absolute value of corresponding generalization errors $e_\mathrm{R}$, $e_\mathrm{P}$ and $e_\mathrm{D}$ at 1000 equispaced points from $-\pi$ to $\pi$.

Next, consider approximating the solution to the Poisson problem (28) with a PINN $u_\mathrm{P}(x;\theta)$. The loss function is

$$\mathcal{L}^\epsilon(\theta) = \frac{1}{N_c} \sum_{i=1}^{N_c} \frac{1}{2} \left( \frac{d^2}{dx^2} u_\mathrm{P}(x_i;\theta) + \left( 4\sin(2x_i) + \frac{1}{\epsilon}\sin(x_i/\epsilon) \right) \right)^2 + \frac{1}{2}\left( [u_\mathrm{P}(-\pi;\theta)]^2 + [u_\mathrm{P}(\pi;\theta)]^2 \right). \quad (31)$$

For $\epsilon = 1/32$ and same neural network as before, we train with $N_c = 1024$ equispaced collocation points for ten independent trials. The training schedule consists of $10,000$, $10,000$, $20,000$, and $20,000$ iterations of Adam with learning rates of $\eta = 10^{-4}$, $\eta = 10^{-5}$, $\eta = 10^{-6}$, and $\eta = 10^{-7}$, respectively, and then L-BFGS for a few thousand iterations, on average.

Figures 4(a) and (b) show the resulting approximations and the generalization errors of the neural networks after training. Taking again $\mathbb{E}\, u_\mathrm{P}(x;\theta)$ to be the network averaged over all the trials, the maximum generalization error is

$$\max_{x_i} |e_\mathrm{P}| := \max_{x_i} |u^\epsilon(x_i) - \mathbb{E}\, u_\mathrm{P}(x_i;\theta)| \approx 7.20 \cdot 10^{-2},$$

while the average variance across the interval is about $1.12 \cdot 10^{-1}$.

Finally, consider the PINN approximation $u_\mathrm{D}(x;\theta)$ of the Darcy problem (29), where the loss function is

$$\mathcal{L}^\epsilon(\theta) = \frac{1}{N_c} \sum_{i=1}^{N_c} \frac{1}{2} \left[ \frac{d}{dx} \left( a(x_i/\epsilon) \frac{d}{dx} u_\mathrm{D}(x_i;\theta) \right) + f^\epsilon(x_i) \right]^2 + \frac{1}{2} \left( [u_\mathrm{D}(-\pi;\theta)]^2 + [u_\mathrm{D}(\pi;\theta)]^2 \right). \quad (32)$$

Once again, $\epsilon = 1/32$, and the network architecture and number of collocation points is the same as the Poisson case. The training schedule consists of $10,000$, $10,000$, $20,000$, and $20,000$ iterations of Adam with

| $\epsilon$ | $\|e_{\mathrm{P}}\|_{L^\infty(\Omega)}$ | $\|\mathrm{Var}_{\mathrm{P}}\|_{L^\infty(\Omega)}$ | $\|e_{\mathrm{P}}\|_{L^\infty(\partial\Omega)}$ | $\|\mathrm{Var}_{\mathrm{P}}\|_{L^\infty(\partial\Omega)}$ |
|---|---|---|---|---|
| $1/64$ | $1.53 \cdot 10^{-1}$ | $5.42 \cdot 10^{-2}$ | $4.21 \cdot 10^{-3}$ | $1.67 \cdot 10^{-5}$ |
| $1/128$ | $1.60 \cdot 10^{-1}$ | $9.83 \cdot 10^{-2}$ | $4.00 \cdot 10^{-3}$ | $5.26 \cdot 10^{-5}$ |

Table 1: $\infty$-norm of the generalization error for the Poisson PINN throughout the domain $\Omega = [-\pi, \pi]$ and on the domain boundary $\partial\Omega$ for $\epsilon \in \{1/64, 1/128\}$. Also shown are the maximum variances in the generalization errors, both in $\Omega$ and on the boundary $\partial\Omega$.

learning rates of $\eta = 10^{-3}$, $\eta = 10^{-4}$, $\eta = 10^{-5}$, and $\eta = 10^{-6}$, respectively, and then L-BFGS for a few thousand iterations, on average. The maximum generalization error

$$\max_{x_i} |e_{\mathrm{D}}| := \max_{x_i} |u^\epsilon(x_i) - \mathbb{E}\, u_{\mathrm{D}}(x_i; \theta)| \approx 4.31 \cdot 10^{-1},$$

which is about 40%, and the average variance across the interval is nearly as large–about $2.83 \cdot 10^{-1}$. From Figures 4(a) and (b) it can be seen that the error contains both low and high frequency components.

### 4.3   Results for decreasing $\epsilon$

We now consider the results for the three different neural network approximations to (27) at the values $\epsilon = 1/64$ and $\epsilon = 1/128$.

Consider first the case of simple $L^2$ regression (i.e. minimizing (30)). For both values of $\epsilon$, $N = \lceil 2\pi/(\epsilon/2) \rceil$ equispaced training points are used for a network with four hidden layers, each of width $d = 40$. Ten independent trials are conducted where the network is trained with the Adam optimizer for 20,000 and 10,000 iterations at a learning rate of $\eta = 10^{-3}$ and $\eta = 10^{-4}$, respectively. The generalization error is tested at 1000 equispaced points in $[-\pi, \pi]$.

The results (not shown) are consistent with the frequency principle, i.e. the known spectral bias of neural networks in the regression setting Xu et al. (2020). The $\infty$-norm of the average generalization error is about $1.71 \cdot 10^{-2}$ and $8.88 \cdot 10^{-3}$ for $\epsilon = 1/64$ and $1/128$, respectively, while in both cases the maximum pointwise variance is relatively small, on the order of $10^{-6}$. The Fourier transform of the error in both cases is essentially zero everywhere except at the $1/\epsilon$ frequency, as expected. The techniques proposed in Yang et al. (2021) for example could be used to mitigate this issue in this setting, if desired.

For the Poisson and Darcy PINNs, we consider a neural network with eight hidden layers, each of width $d = 40$. The training schedule consists of 2000 iterations of Adam with a learning rate of $\eta = 10^{-6}$, 5000 iterations with $\eta = 10^{-7}$, 10000 iterations with $\eta = 10^{-8}$ and then L-BFGS. The pointwise generalization errors for the two cases $\epsilon = 1/64$ and $\epsilon = 1/128$ are shown in Figures 5 and 6 in Appendix F for the Poisson and Darcy PINNs, respectively. The $\infty$-norms of the average generalization error for each $\epsilon$ case are reported in Tables 1; and 2 for the Poisson and Darcy PINNs, respectively. Also reported are the maximum average errors on the boundary $\partial\Omega = \{-\pi, \pi\}$, as well as the $\infty$-norm of the variance in the generalization error throughout $\Omega$ and on $\partial\Omega$.

Neither PINN does particularly well, as expected from the theory developed here for the Darcy case and the analysis in Wang et al. (2021b; 2022) for the Poisson case. Although there are some similarities in the error profiles–for example, the variance over the independent trials becomes larger for both PINN cases as $\epsilon$ decreases–there are some important differences to highlight as well.

For the Poisson PINN, the average generalization error is approximately the same at both values of $\epsilon$, and in particular, the PINN satisfies the homogeneous Dirichlet BC to about three digits of accuracy for each $\epsilon$. To ensure this, we had to set the boundary weight $\lambda_{\mathrm{BC}} = 10$ (this value was kept consistent for all the cases listed in Tables 1 and 2). In contrast, the average error in the Darcy case increases for decreasing $\epsilon$, and in particular, the error at the boundaries increases. These results are consistent with the discussion in Section 3.2, i.e. that the oscillatory coefficient within the differential operator for the Darcy PINN worsens the imbalance between the two loss terms $\mathcal{L}_{\mathrm{PDE}}$ and $\mathcal{L}_{\mathrm{BC}}$ already known to exist for Poisson PINNs.

| $\epsilon$ | $\|e_{\mathrm{D}}\|_{L^\infty(\Omega)}$ | $\|\mathrm{Var}_{\mathrm{D}}\|_{L^\infty(\Omega)}$ | $\|e_{\mathrm{D}}\|_{L^\infty(\partial\Omega)}$ | $\|\mathrm{Var}_{\mathrm{D}}\|_{L^\infty(\partial\Omega)}$ |
|---|---|---|---|---|
| 1/64 | $1.01 \cdot 10^{-1}$ | $8.43 \cdot 10^{-2}$ | $7.84 \cdot 10^{-2}$ | $8.34 \cdot 10^{-2}$ |
| 1/128 | $3.80 \cdot 10^{-1}$ | $3.06 \cdot 10^{-1}$ | $3.11 \cdot 10^{-1}$ | $2.99 \cdot 10^{-1}$ |

Table 2: $\infty$-norm of the generalization error for the Darcy PINN throughout the domain $\Omega = [-\pi, \pi]$ and on the domain boundary $\partial\Omega$ for $\epsilon \in \{1/64, 1/128\}$. Also shown are the maximum variances in the generalization errors, both in $\Omega$ and on the boundary $\partial\Omega$.

### 4.4 Rescaling the loss function

As mentioned in Section 3.2, it is reasonable to try mollify the imbalance between the two parts of the Darcy PINN loss function by rescaling the PDE residual portion $\mathcal{L}_{\mathrm{PDE}}$ by $\epsilon^2$, as this cancels the $1/\epsilon^2$ factor in the NTK matrix $K^\epsilon_{uu}$ subblock. Here we test this idea at a sequence of $\epsilon$ values.

We consider neural networks with three hidden layers each of width $d = 70$. As before, we use hyperbolic tangent activation functions with $N_c = 1024$ collocation points. The training schedule consists of ten thousands iterations each of the learning rates $\eta = 10^{-3}$, $10^{-4}$, $10^{-5}$ and $10^{-6}$, and then, on average a few thousand iterations of L-BFGS. As before, we conduct ten independent trials and then average the results.

In particular we consider learning the solution function

$$u^\epsilon(x) = e^{-(x-1)^2} + \epsilon \sin(x/\epsilon) + c_1 x + c_2$$

where $c_1$ and $c_2$ are chosen such that $u^\epsilon$ satisfies homogeneous Dirichlet boundary conditions on the domain $[-\pi, \pi]$; with

$$a^\epsilon(x) = 1.1 + \cos(x/\epsilon)\cos(2x/\epsilon),$$

we set the forcing function $f^\epsilon$ in the Darcy problem such that $u^\epsilon$ indeed solves

$$-\frac{d}{dx}\left(a^\epsilon(x)\frac{d}{dx}u^\epsilon(x)\right) = f^\epsilon(x). \tag{33}$$

The pointwise generalization errors for three values of $\epsilon$, as well as the magnitude of their Fourier coefficents, are shown in Figure 7 in Appendix F. Compared to the Darcy PINNs without rescaling, the homogeneous Dirichlet conditions are much better satisfied. For the $\epsilon = 1/36$, the average error on the boundary is on the order of $10^{-3}$, consistent with the Poisson cases in the previous section. For $\epsilon = 1/72$ and $\epsilon = 1/144$, the accuracy is lower (on the order of $10^{-1}$), but the variance is quite low, in contrast to the huge variance observed in the unscaled Darcy PINNs. In each case, the error in Fourier space has both a low frequency peak concentrated in the first few wavenumbers and a peak at the $1/\epsilon$ frequency; on average the errors in these peaks increase for decreasing $\epsilon$. In particular, at the smallest value $\epsilon = 1/144$, an additional low frequency peak emerges around $k = 16$.

As discussed extensively in Wang et al. (2022) and mentioned in Section 3.2, both the magnitude and the distribution of the eigenvalues of the NTK matrix subblocks are closely connected with PINN performance. As shown in Figure 3, there is a large discrepancy between the principal eigenvalue of the $K^\epsilon_{uu}$ subblock and the rest of the spectrum, which is a characteristic sign that the gradient flow dynamics (11) are stiff. Although the rescaling introduced by setting $\lambda_{\mathrm{PDE}} = \epsilon^2$ changes the magnitude of the eigenvalues of the $K^\epsilon_{uu}$ subblock, it of course does not change their distribution, and we attribute the increasing errors at decreasing $\epsilon$ values to the discrepancy in the spectrum.

## 5 Conclusions

Physics informed neural networks have demonstrated success in a wide variety of problems in scientific computing (Karniadakis et al., 2021), however, they can sometimes struggle to approximate solutions to differential equations with multiscale features. Athough PINN convergence theory guarantees that minimizers of (regularized) loss functions can in principle well approximate solutions to elliptic problems, in practice,

achieving those minimizers is difficult for boundary value problems with highly oscillatory coefficients present in the differential operator.

We show that for a class of linear, multiscale elliptic equations in divergence form, the Frobenius norm of the neural tangent kernel matrix associated to the PINN becomes unbounded as the characteristic wavelength of the oscillations vanishes. Numerical examples illustrate that during training, the ordinary differential equation that governs the evolution of the PDE residuals becomes increasingly stiff as $\epsilon \downarrow 0$, translating to poor PINN training behavior.

The present work considers standard, fully connected neural network architectures; it may be possible in future research to develop alternative PINN architectures that are specifically adapted to the problem. For example, different activation functions with problem-tuned inductive biases can be used, as suggested in Ziyin et al. (2020) for periodic functions. Li et al. (2020a) also proposed smooth and localized activation functions for multiscale elliptic problems in the context of the Deep Ritz method Yu et al. (2018). In this context, it may be beneficial to develop a theory for the Neural Tangent Kernel matrix for energy-minimization based neural network methods for solving multiscale PDEs, as the derivative of the oscillatory coefficient $a^\epsilon$ does not appear, in contrast to PINNs.

Future work may also benefit from more sophisticated learning techniques beyond (stochastic) gradient descent, for example genetic algorithms (Mirjalili, 2019). Another possibility is to employ continuation methods (Allgower & Georg, 2003), i.e. "curriculum regularization" (Krishnapriyan et al., 2021).

Recent results establishing a "frequency principle" in machine learning motivated this study; during training, neural networks tend to learn their target functions from low to high frequency (Luo et al., 2021). See also the work by Wang et al. (2021b) for spectral analysis of physics informed neural networks applied to Poisson problems with multiscale forcing. An interesting avenue for future study is to characterize the spectral behavior during training of deep learning methods for learning operators; see for example (Li et al., 2020b; Khoo et al., 2021; Bhattacharya et al., 2021; Lu et al., 2021a; Zhang et al., 2022). In the context of the present work, this would be the nonlinear map from the problem data $a^\epsilon$ and $f$ in (3) to the solution $u^\epsilon$. Finally, we note many other novel neural network approaches to multiscale elliptic problems not discussed here have recently been proposed, for example (Han & Lee, 2023; Leung et al., 2022; Fabian et al., 2022). We hope some of the results presented here might inspire insight into these exciting approaches.

### Acknowledgments

The authors thank Andrea Bertozzi for organizing the UCLA Computational and Applied Mathematics REU site where much of this work took place. The authors also thank Chris Anderson and Michael Murray for helpful discussions.

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

## A  Description of PINN approximation for multiscale Poisson problem

Here we give a more complete description of the example from Section 2 of a PINN approximation to the Poisson boundary value problem (4) with homogeneous Dirichlet boundary conditions.

The PINN solution is computed with a fully connected neural network with four hidden layers of sixty nodes each and hyperbolic tangent activation functions. The training consisted of $100$, $1,000$, $15,000$, $30,000$ and $10,000$ iterations with the Adam optimizer with learning rates of $\eta = 10^{-2}$, $\eta = 10^{-3}$, $\eta = 10^{-4}$, $\eta = 10^{-5}$, and $\eta = 10^{-6}$, respectively. We note, however, using a constant value of $\eta = 10^{-3}$ throughout training produced results qualitatively similar to those in Figure 1; i.e. the observed frequency principle was fairly robust to changes in the learning rate. As mentioned in the description in Section 2, for legibility, the results shown in Figure 1 are that of the rolling average (of 50 values) to the trajectories of the error in the Fourier coefficients.

Finally, we mention that at the end of the training process, the $\infty$-norm error between the true solution and the neural network approximation is about $1.26 \cdot 10^{-2}$.

## B  Proof of Lemma 3.1

*Proof.* Computing first the entries of the gradient of the loss function (8) gives

$$\frac{\partial \mathcal{L}^\epsilon}{\partial \theta_l} = \frac{1}{N_c} \sum_{j=1}^{N_c} r_{\text{pde}}(x_j; \theta) \frac{\partial}{\partial \theta_l} L^\epsilon u(x_j; \theta) + \frac{\lambda}{N_b} \sum_{j=1}^{N_b} r_{\text{b}}(s_j; \theta) \frac{\partial}{\partial \theta_l} u(s_j; \theta). \tag{34}$$

Next, compute

$$\frac{d}{dt} r_{\text{pde}}(x_i; \theta(t)) = \frac{d}{dt} \left( L^\epsilon u(x_i; \theta(t)) - f(x_i) \right) = \sum_{l=1}^{N_p} \frac{\partial}{\partial \theta_l} L^\epsilon u(x_i; \theta) \frac{d\theta_l}{dt}$$

$$= -\sum_{l=1}^{N_p} \frac{\partial}{\partial \theta_l} L^\epsilon u(x_i; \theta) \frac{\partial \mathcal{L}^\epsilon}{\partial \theta_l}, \tag{35}$$

where the final equality comes from the assumption of gradient flow dynamics (11). A similar computation gives

$$\frac{d}{dt} r_{\text{b}}(s_i; \theta(t)) = \frac{d}{dt} \left( u(s_i; \theta(t)) - g(s_i) \right) = \sum_{l=1}^{N_p} \frac{\partial}{\partial \theta_l} u(s_i; \theta) \frac{d\theta_l}{dt}$$

$$= -\sum_{l=1}^{N_p} \frac{\partial}{\partial \theta_l} u(s_i; \theta) \frac{\partial \mathcal{L}^\epsilon}{\partial \theta_l}, \tag{36}$$

Inserting (34) into (35) and rearranging the sums gives

$$\frac{d}{dt} r_{\text{pde}}(x_i; \theta(t)) = -\Big( \sum_{j=1}^{N_c} \Big[ \frac{1}{N_c} \sum_{l=1}^{N_p} \frac{\partial}{\partial \theta_l} L^\epsilon u(x_i; \theta) \frac{\partial}{\partial \theta_l} L^\epsilon u(x_j; \theta) \Big] r_{\text{pde}}(x_j; \theta)$$

$$+ \sum_{j=1}^{N_b} \Big[ \frac{\lambda}{N_b} \sum_{l=1}^{N_p} \frac{\partial}{\partial \theta_l} L^\epsilon u(x_i; \theta) \frac{\partial}{\partial \theta_l} u(s_j; \theta) \Big] r_{\text{b}}(s_j; \theta) \Big),$$

while inserting (34) into (36) similarly results in

$$\frac{d}{dt} r_{\text{b}}(s_i; \theta(t)) = -\Big( \sum_{j=1}^{N_c} \Big[ \frac{1}{N_c} \sum_{l=1}^{N_p} \frac{\partial}{\partial \theta_l} u(s_i; \theta) \frac{\partial}{\partial \theta_l} L^\epsilon u(x_j; \theta) \Big] r_{\text{pde}}(x_j; \theta)$$

$$+ \sum_{j=1}^{N_b} \Big[ \frac{\lambda}{N_b} \sum_{l=1}^{N_p} \frac{\partial}{\partial \theta_l} u(s_i; \theta) \frac{\partial}{\partial \theta_l} u(s_j; \theta) \Big] r_{\text{b}}(s_j; \theta) \Big)$$

as desired.

□

## C   Proof of Theorem 3.3

*Proof.* From (25) we note that $W^{(1)} \in \mathbb{R}^{d_1 \times 1}$, $W^{(l)} \in \mathbb{R}^{d_l \times d_{l-1}}$ for $l = 2, \ldots, \Lambda$, and $W^{(\Lambda+1)} \in \mathbb{R}^{1 \times d_\Lambda}$, while $b^{(l)} \in \mathbb{R}^{d_l}$ for $l = 1, \ldots, \Lambda$ and $b^{(\Lambda+1)} \in \mathbb{R}$. The fully connected neural network (25) can then be written as

$$u(x; \theta) = \sum_{k_\Lambda=1}^{d_\Lambda} W_{k_\Lambda}^{(\Lambda+1)} u_{k_\Lambda}^{(\Lambda)}(x) + b^{(\Lambda+1)},$$

$$u_{k_\Lambda}^{(\Lambda)}(x) = \sigma\Big( \sum_{k_{\Lambda-1}=1}^{d_{\Lambda-1}} W_{k_\Lambda k_{\Lambda-1}}^{(\Lambda)} u_{k_{\Lambda-1}}^{(\Lambda-1)}(x) + b_{k_\Lambda}^{(\Lambda)} \Big), \qquad 1 \leq k_\Lambda \leq d_\Lambda,$$

$$u_{k_{\Lambda-1}}^{(\Lambda-1)}(x) = \sigma\Big( \sum_{k_{\Lambda-2}=1}^{d_{\Lambda-2}} W_{k_{\Lambda-1} k_{\Lambda-2}}^{(\Lambda-1)} u_{k_{\Lambda-2}}^{(\Lambda-2)}(x) + b_{k_{\Lambda-1}}^{(\Lambda-1)} \Big), \qquad 1 \leq k_{\Lambda-1} \leq d_{\Lambda-1},$$

$$\vdots$$

$$u_{k_2}^{(2)}(x) = \sigma\Big( \sum_{k_1=1}^{d_1} W_{k_2 k_1}^{(2)} u_{k_1}^{(1)}(x) + b_{k_2}^{(2)} \Big), \qquad 1 \leq k_2 \leq d_2$$

$$u_{k_1}^{(1)}(x) = \sigma\Big( W_{k_1}^{(1)} x + b_{k_1}^{(1)} \Big), \qquad 1 \leq k_1 \leq d_1.$$

By linearity, note that

$$\frac{\partial}{\partial W_\gamma^{(\Lambda+1)}} L^\epsilon u(x; \theta) = L^\epsilon u_\gamma^{(\Lambda)}(x) \tag{37}$$

for any $1 \leq \gamma \leq d_\Lambda$.

As before, consider a diagonal entry in the $K_{uu}^\epsilon$ subblock of the NTK matrix (14)

$$(K_{uu}^\epsilon)_{\alpha\alpha}(t) = \frac{1}{N_c} \sum_{\gamma=1}^{N_p} \frac{\partial}{\partial \theta_\gamma} L^\epsilon u(x_\alpha; \theta(t)) \frac{\partial}{\partial \theta_\gamma} L^\epsilon u(x_\alpha; \theta(t)).$$

By (37), a sufficient condition for the Frobenius norm of $K^\epsilon$ to blow-up, then, is for

$$\lim_{\epsilon \downarrow 0} \big( L^\epsilon u_\gamma^{(\Lambda)}(x_\alpha) \big)^2 = \infty$$

for some collocation point $x_\alpha$, some $1 \leq \gamma \leq d_\Lambda$, and $t \in \Upsilon$ (here the dependence of $u_\gamma^{(\Lambda)}$ on $t$ and $\theta$ is implied). Since

$$\big( L^\epsilon u_\gamma^{(\Lambda)}(x_\alpha) \big)^2 = \Big[ \Big( a(x_\alpha/\epsilon) \frac{d^2}{dx^2} u_\gamma^{(\Lambda)}(x_\alpha) \Big)^2 + \frac{2}{\epsilon} \Big( a(x_\alpha/\epsilon) \frac{d^2}{dx^2} u_\gamma^{(\Lambda)}(x_\alpha) \, a'(x_\alpha/\epsilon) \frac{d}{dx} u_\gamma^{(\Lambda)}(x_\alpha) \Big)$$
$$+ \frac{1}{\epsilon^2} \Big( a'(x_\alpha/\epsilon) \frac{d}{dx} u_\gamma^{(\Lambda)}(x_\alpha) \Big)^2 \Big], \tag{38}$$

and since $a$ is a periodic $C^1$ function, the desired result will follow if (i) the second derivative of $u_\gamma^{(\Lambda)}$ is uniformly bounded in $\epsilon$ and (ii) the first derivative is asymptotically larger than $\epsilon$, so that

$$\lim_{\epsilon \downarrow 0} \frac{1}{\epsilon} \Big| \frac{d}{dx} u_\gamma^{(\Lambda)}(x) \Big| = \infty \tag{39}$$

for any $t \in \Upsilon$.

The first derivative is

$$\frac{d}{dx}u_\gamma^{(\Lambda)}(x) = \sigma'\Big(\sum_{k_{\Lambda-1}=1}^{d_{\Lambda-1}} W_{\gamma k_{\Lambda-1}}^{(\Lambda)} u_{k_{\Lambda-1}}^{(\Lambda-1)}(x) + b_{k_\Lambda}^{(\Lambda)}\Big) \sum_{k_{\Lambda-1}=1}^{d_{\Lambda-1}} \Big[ W_{\gamma k_{\Lambda-1}}^{(\Lambda)} \frac{d}{dx} u_{k_{\Lambda-1}}^{(\Lambda-1)}(x)\Big] \tag{40}$$

after applying the chain rule once. Using Einstein summation notation for brevity, (40) simplifies to

$$\frac{d}{dx}u_\gamma^{(\Lambda)}(x) = \sigma'\Big(W_{\gamma k_{\Lambda-1}}^{(\Lambda)} u_{k_{\Lambda-1}}^{(\Lambda-1)}(x) + b_{k_\Lambda}^{(\Lambda)}\Big)\Big[W_{\gamma k_{\Lambda-1}}^{(\Lambda)} \frac{d}{dx} u_{k_{\Lambda-1}}^{(\Lambda-1)}(x)\Big].$$

Continuing with the chain rule (and the Einstein notation), we get

$$\frac{d}{dx}u_\gamma^{(\Lambda)}(x) = \sigma'\Big(W_{\gamma k_{\Lambda-1}}^{(\Lambda)} u_{k_{\Lambda-1}}^{(\Lambda-1)}(x) + b_{k_\Lambda}^{(\Lambda)}\Big) \times$$
$$\Big[W_{\gamma k_{\Lambda-1}}^{(\Lambda)} \sigma'\Big(W_{k_{\Lambda-1}k_{\Lambda-2}}^{(\Lambda-1)} u_{k_{\Lambda-2}}^{(\Lambda-2)}(x) + b_{k_{\Lambda-1}}^{(\Lambda-1)}\Big)\Big[W_{k_{\Lambda-1}k_{\Lambda-2}}^{(\Lambda-1)} \frac{d}{dx} u_{k_{\Lambda-2}}^{(\Lambda-2)}(x)\Big]\Big].$$

From here we can recurse downwards to compute the full derivative as

$$\frac{d}{dx}u_\gamma^{(\Lambda)}(x) = \sigma'\Big(W_{\gamma k_{\Lambda-1}}^{(\Lambda)} u_{k_{\Lambda-1}}^{(\Lambda-1)}(x) + b_{k_\Lambda}^{(\Lambda)}\Big) \times$$
$$\Big[W_{\gamma k_{\Lambda-1}}^{(\Lambda)} \sigma'\Big(W_{k_{\Lambda-1}k_{\Lambda-2}}^{(\Lambda-1)} u_{k_{\Lambda-2}}^{(\Lambda-2)}(x) + b_{k_{\Lambda-1}}^{(\Lambda-1)}\Big)\Big[W_{k_{\Lambda-1}k_{\Lambda-2}}^{(\Lambda-1)} \sigma'\Big(W_{k_{\Lambda-2}k_{\Lambda-3}}^{(\Lambda-2)} u_{k_{\Lambda-3}}^{(\Lambda-3)}(x) + b_{k_{\Lambda-2}}^{(\Lambda-2)}\Big) \times$$
$$\Big[\dots \Big[W_{k_3 k_2}^{(3)} \sigma'\Big(W_{k_2 k_1}^{(2)} u_{k_1}^{(1)}(x) + b_{k_2}^{(2)}\Big)\Big[W_{k_2 k_1}^{(2)} \sigma'\Big(W_{k_1}^{(1)} x + b_{k_1}^{(1)}\Big) W_{k_1}^{(1)}\Big]\Big]\dots\Big]\Big]\Big]. \tag{41}$$

By the uniform boundedness assumption (19) on the network parameters $\theta$, (41) is uniformly bounded in $\epsilon$ from above for any $x \in [a,b]$. As a consequence of the positivity assumption (18), (19) further implies that each instance of $d\sigma/dx$ in (41) is bounded below by a positive number that is independent of $\epsilon$. Note that (41) generally contains sums and products of $d\sigma/dx$ and network weights (entries in the $W^{(l)}$ matrices). For any $t \in \Upsilon$, the weights are asymptotically larger than $\epsilon^{1/\Lambda}$, so that

$$\lim_{\epsilon \downarrow 0}\Big(\frac{\epsilon}{W_{k_1}^{(1)} W_{k_2 k_1}^{(2)} \dots W_{k_{\Lambda-1}k_{\Lambda-2}}^{(\Lambda-1)} W_{k_\Lambda k_{\Lambda-1}}^{(\Lambda)}}\Big) = 0$$

for any combination of indices $1 \le k_1 \le d_1$, $1 \le k_2 \le d_2$, and so on, meaning that (39) indeed holds.

Ergo, as in the proof of Theorem 3.2, as long as $\epsilon$ vanishes monotonically to zero in such a way that $a'(x_\alpha/\epsilon) \neq 0$ any $\epsilon$, (38) limits to positive infinity as desired.

$\square$

## D Proof of Corollary 3.4

*Proof.* From elementary linear algebra, we know that the Frobenius norm of the square matrix $K_{uu}^\epsilon(t)$ equals the Euclidean norm of its singular values, and because $K_{uu}^\epsilon(t)$ is a symmetric matrix, its singular values equal the absolute value of its (real valued) eigenvalues. If $\{\lambda_i^\epsilon(t)\}_{i=1}^{N_c}$ denotes the eigenvalues, then

$$\|K_{uu}^\epsilon(t)\|_F = \Big(\sum_{i=1}^{N_c}(\lambda_i^\epsilon(t))^2\Big)^{1/2} \le \sqrt{N_c}\,\rho(K_{uu}^\epsilon(t))$$

giving the result. $\square$

## E Boundary value problem forcing functions

The right-hand side forcing function $f^\epsilon(x)$ for the Darcy BVP (29) with homogenenous Dirichlet boundary conditions is given by $g^\epsilon(x)/h^\epsilon(x)$, where

$$g^\epsilon(x) = 10\Big[20\,\pi + 168\,\pi\epsilon\cos(x)\sin(x) - 20\Big(2\pi - 4\pi\cos^2(x) + \epsilon\sin(\pi/\epsilon)\Big)\cos(x/\epsilon)$$
$$+ \Big(21\pi + 160\,\pi\epsilon\cos(x)\sin(x)\Big)\sin(x/\epsilon)\Big],$$

and

$$h^\epsilon(x) = 400\,\pi\epsilon \sin^2(x/\epsilon) + 840\,\pi\epsilon \sin(x/\epsilon) + 441\,\pi\epsilon.$$

# F    Error plots for decreasing $\epsilon$

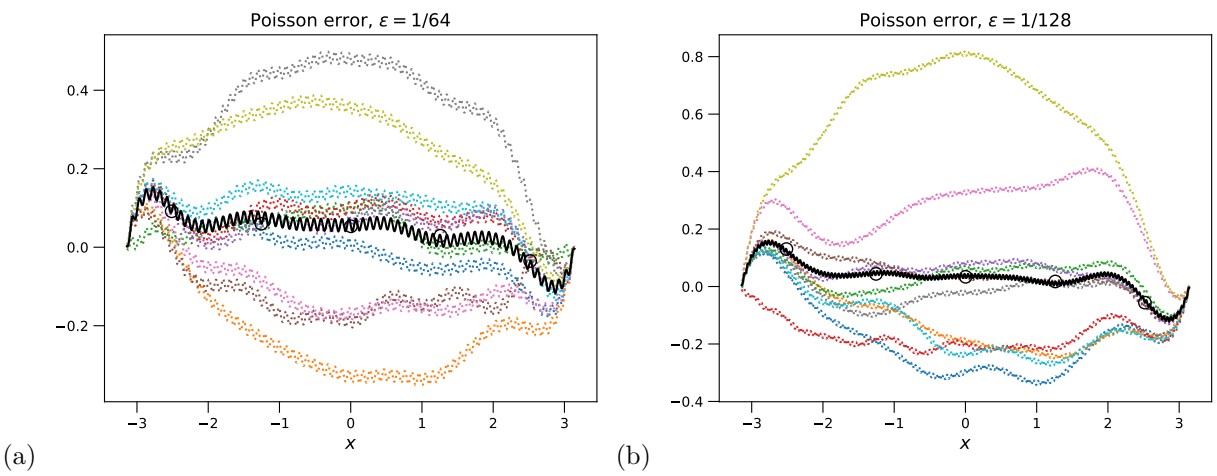

(a)                                                          (b)

Figure 5:   Pointwise generalization errors for the Poisson PINN (31). The dotted lines indicate independent trial runs, while the solid line with circle markers indicates the mean.

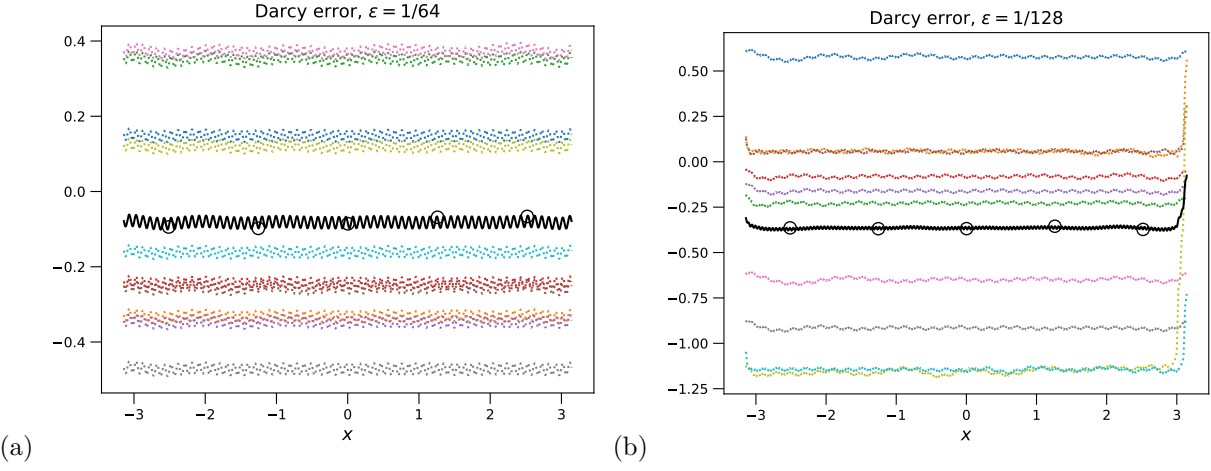

(a)                                                          (b)

Figure 6:   Pointwise generalization errors for the Darcy PINN (32). The dotted lines indicate independent trial runs, while the solid line with circle markers indicates the mean.

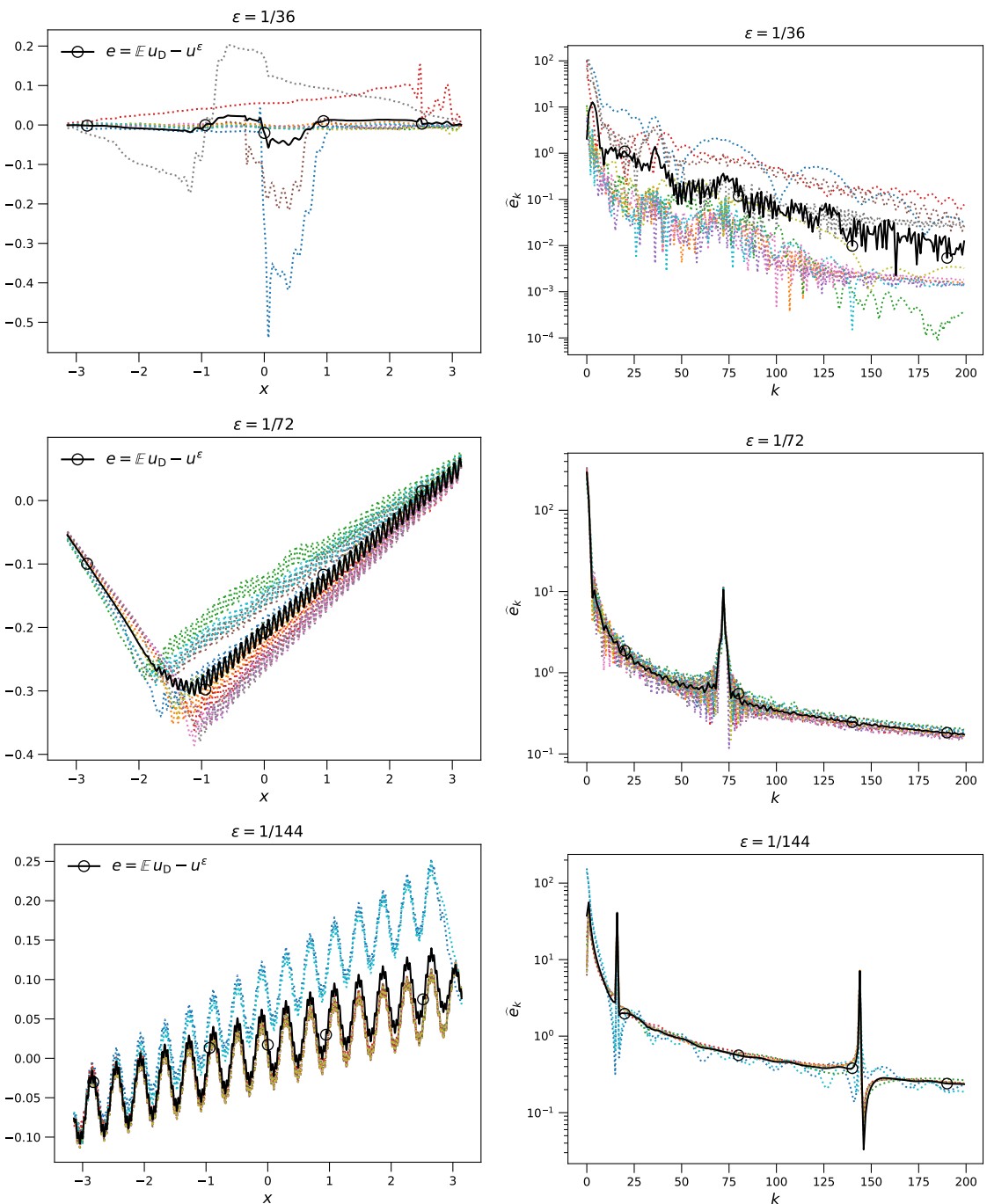

Figure 7: Results at three different $\epsilon$ values for the Darcy PINN associated to (33) with the PDE residual part of the loss function weighted by $\lambda_{\mathrm{PDE}} = \epsilon^2$. The left columns display the pointwise generalization error on $[-\pi, \pi]$, and the right columns display the magnitude of the first two hundred Fourier coefficients of the error. The dotted lines indicate independent trial runs, while the solid line with circle markers indicates the mean.

