# OpenReview forum: "Physics informed neural networks for elliptic equations with oscillatory differential operators"
_TMLR — Accepted by TMLR_

### Review · Reviewer_37N7 · 2023-08-04

**Summary Of Contributions:**

This work proves that the NTK of PINN increases as $\epsilon^{-2}$ where $\epsilon^{-1}$ is the frequency of the highest mode in the differential equation

**Audience:**

Yes

**Claims And Evidence:**

No

**Requested Changes:**

Please answer the weaknesses I raised and adapt the draft accordingly.

**Strengths And Weaknesses:**

The strength is the novelty of the idea that studying NTK can help us understand PINN and that the theory feels sufficiently sound.

There are a few weaknesses I think this work needs to address

1. The main result states that the NTK change by the order ($\epsilon^{-2}$). But I am a little confused by what this statement means. Does it mean that before and after training, the NTK will change by order $\epsilon^{-2}$? Or it simply means that before training, the NTK will change because of the change in the loss function of order $\epsilon^{-1}$

2. The argument that NTK changes as $\epsilon^{-2}$ "implies that... training the neural network with gradient descent based methods to achieve an accurate approximation of the solution to the PDE becomes increasingly difficult" is tenuous. I do not see why the claim is implied, the authors need to explain this in detail. After all, I think this claim cannot be supported by the theory of the paper. NTK is only relevant to the case when the learning rate is vanishingly small and is not relevant when the learning rate is large. In fact, this work might imply that it is better to use a large learning rate, a point that is not discussed in the main text

3. it seems that the problem of training identified in this paper already has a solution. The authors should discuss the relevant work of https://arxiv.org/abs/2006.08195.  The problem with PINN is due to the fact that the assumed model activation function is monotonic (Eq. 18), whereas learning the right solution requires a periodic function. It seems to the audience that the problem simply comes from the fact that the inductive bias of the model does not match the problem. Including a discussion of the result in https://arxiv.org/abs/2006.08195 and performing numerical experiments to either validate or disprove what I state above will greatly improve the manuscript

---

> ### Author Response · Authors · 2023-08-31
> **Author response**
>
> We thank the referee for the detailed review and helpful suggestions for improving the manuscript. Please refer to the "response3.pdf" file uploaded in the supplementary materials.
>
> The paper has been revised in response to reviewer suggestions and to correct minor errors. Significant changes are highlighted in blue.
> Further, our response to the questions and suggestions are detailed below.
>
>   -- Comment 1 response --
>
> This was not well explained in the text, and the confusion you point out may be linked to the question raised in Comment 2 below.
> The analysis in the paper is inspired by asymptotic homogenization theory, where one studies a *sequence of problems*, indexed by $\epsilon$, to try and understand their limiting behavior as $\epsilon$ vanishes.
>
> So, for a *fixed* $\epsilon$ (say $\epsilon_1$), the NTK matrix simply changes as a function of time $t$ according to its definition in Eq. (14) (here we are assuming the network parameters evolve according to a continuous time gradient flow, as in Eq. (11); see the additional discussion on this point below). Suppose then that one measures the Frobenius norm at time $T$ to be some number $C$:
> $$ \| \|  K^{\epsilon_1}(T) \| \|_F = C. $$
> If one then considers again the same problem but with $\epsilon_2 = \epsilon_1/2$. Then the main result simply says that one can expect
> $$
> \| \| K^{\epsilon_2}(T) \| \|_F \approx 4 C.
> $$
> The punchline is that problems with large scale separation can be expected to have larger NTK matrices than those with moderate or mild scale separation.
>
>   -- Comment 2 response --
>
> Thank you for raising an issue with our claim that "training...becomes increasingly difficult", as this critical argument was not properly developed in the previous version of the paper. We have updated the text to include a discussion of this point in some detail; please see the discussion in the new Section 3.1, as well as the additional numerical experiments in Sections 4.3 and 4.4 in support of our main claim.
>
> The theory developed in the paper builds on the previous analyses of PINN training from Wang et al. (SIAM J. Sci. Comput., 2021), Wang et al. (Comput. Meth. Appl. Mech. Engr., 2021) and Wang et al. (J. Comput. Phys., 2022) in which PINN training was analyzed from the lens of a vanishingly small learning rate. This is natural to consider, since the standard optimizers used for PINNs (such as Adam and L-BFGS) are only guaranteed to find local, rather than global minimizers. A relatively large learning rate risks missing out on local optima
> and can be unstable from the point of classical numerical analysis; see section 4.2 of Wang et al. (SIAM J. Sci.\ Comput.) for a nice discussion of this latter point.
>
>   -- Comment 3 response --
>
> Thank you for this suggestion to improve the manuscript. We included a comment in the Conclusions section that highlights the possibility of selecting alternative, problem-specific activation functions that can improve the network's inductive biases.
>
> The theory in the paper characterizes PINN performance for divergence-form elliptic problems with general Dirichlet conditions; the examples in section 4.2 were chosen to illustrate poor performance (for equations of the form (28)) even on relatively simple, one-dimensional problems. For simplicity the boundary conditions were taken to be homogeneous Dirichlet, which indeed implies in this case that the solution is periodic (it equals zero at both endpoints).
>
> Hence, the activation function proposed in Ziyin et al. (NeurIPS 2020) may indeed improve the PINN performance equations of the form (28) ("Darcy" form) for this particular case; they may also further improve the results for the regression problem and Poisson problem, although we note that reasonable approximations for these cases were already achieved with the monotonic, hyperbolic tangent activation function.
>
> Nevertheless, one would hope that PINN approximation of Darcy form equations can succeed in more general situations where one doesn't have the relative simplicity of periodic boundary conditions. Also, in practice one can encounter multiscale equations of the form (3) where the microscale is not periodic (i.e. the coefficient $a^{\epsilon}(x)$ is not a periodic function); in this case alternative activation functions to the ones proposed in Ziyin et al. would be needed. For this reason we focus here on the most commonly used PINN activation functions.
>
> Finally, we also note that the activation functions proposed in Ziyin et al. appear to be designed for extrapolation of temporally varying functions, which is fundamentally different that the boundary value problems considered here.

---

> > ### Comment · Reviewer_37N7 · 2023-09-07
> > **reply**
> >
> > Thanks for the response. I think the response answered my questions

---

### Review · Reviewer_DENr · 2023-08-09

**Summary Of Contributions:**

This paper investigates the pitfalls of Physics informed neural networks for solving multiscale elliptic PDEs where the solutions exhibit oscillatory behavior. The authors show that when the coefficient of the elliptic operator contains frequency at the order of $1/\epsilon$, the Frobenius norm of the NTK matrix associated to the loss function scales like $1/\epsilon$, whereby claiming that training the networks with gradient-descent based methods becomes increasingly difficult for highly oscillatory problems. Numerical experiments are provided to illustrate the stiffness of the problem.

**Audience:**

No

**Claims And Evidence:**

Yes

**Requested Changes:**

I do not have many comments on the possible changes. But I do want to add that it is certainly an interesting and important problem to design efficient deep learning methods for solving multiscale PDEs. Unfortunately, the present paper only points out the issue of PINNs and could be substantially improved if the authors were to propose new algorithm/methodology that can solve/alleviate the issue.

**Strengths And Weaknesses:**

Strength: No

Weaknesses: The major weakness of the paper is a lack of sufficient novelty. The fact that GD is difficult to learn oscillatory functions with high frequency is known in the literature; see Xu et al. 2020 for regression setting and other references mentioned by the authors. In the PDE setting, Wang et al. 2021 and Wang et. al 2022 also studied the convergence of training process as well as the spectral bias from an NTK perspective. The only difference between the present paper and these earlier works lies in the investigation of the blow-up of the norm of the NTK matrix as shown in Theorem 3.2 and Theorem 3.3. The proofs of both results follow from straightforward calculations of the NTK matrix and do not seem to require new technicalities or ideas.

---

> ### Author Response · Authors · 2023-08-31
> **Author response**
>
> We thank the referee for the reviewing the manuscript and for emphasizing the importance of developing deep learning methods for multiscale partial differential equations.
>
>    -- Comment: "The major weakness of the paper is a sufficient lack of novelty ..." --
>
> Thank you for pointing out a deficiency in the original draft of the manuscript, namely that there is no discussion of the similaries and differences between the present work and previous analyses of PINN failure modes.
>
> The present work is certainly inspired by the series of papers by Wang et al. (SIAM J. Sci. Comp. 2021; Comput. Meth. Appl. Mech. Engr. 2021;  J. Comput. Phys., 2022) who point out that, fundamentally, training PINNs involves optimizing a multi-objective loss functional;
> the PDE residual and boundary conditions residual (for stationary problems) need to simultaneously be minimized. Broadly speaking, PINNs can fail to train whenever these two terms in the loss function are imbalanced.
>
> Previous analyses of such imbalances have focused on the case of Poisson-type boundary value problems where the Laplacian was the differential operator (or, in one dimension, the second derivative). Any multiscale features in the problem originated from oscillatory forcing functions, or "right-hand sides". In contrast, in the present work the multiscale nature of the problems considered originates from
> an oscillatory function *within in the differential operator itself* (which we term "Darcy-type" equations).
>
> The differences between the two cases can be considerable, both in theory and practice, which is a key takeaway of the present work. For example, for the Darcy problem, the spectral radius of the $K_{uu}$ matrix subblock scales like $1/\epsilon^2$, as pointed out in the new Corollary 3.4 and illustrated in Figure 2. For multiscale Poisson problems however, this matrix subblock is independent of $\epsilon$, since the oscillatory forcing function $f$ is independent of the network parameters $\theta$.
>
> We added a new section to the paper that discusses the theoretical differences in more detail; please refer to Section 3.2.  We also expanded the numerical experiments to highlight the difference that can arise in practice; please refer to the new Section 4.3.
>
>
>    -- Comment: "Unfortunately, the present paper only points out the issue of PINNs and could be substantially improved
> if the authors were to propose new algorithm/methodology that can solve/alleviate the issue." --
>
> We respectfully defer to alternative approaches for solving Darcy-type equations with neural network based approaches found e.g. in Han and Lee (Multiscale Model. Simul., 2023) or Leung et al. (J. Comput. Phys., 2022). As these involve considerable developments beyond the standard PINN methodology (in the latter work) or abandoning it entirely (in the former), we focus in the present work on shedding light on the strengths and weaknesses of existing PINNs techniques, and in particular explaining *why* those weaknesses are present for the class of equations considered here.

---

### Review · Reviewer_c8m4 · 2023-08-17

**Summary Of Contributions:**

The authors consider the training dynamics of physics-informed neural networks. More specifically, they show that for elliptic equation (in flux-conservative form) with strongly oscillating coefficients $a_{\epsilon}(x) := f(x\big/\epsilon),$ $f(y) = f(y+1),$ $f(y)\in C^{1}([0, 1])$, neural tangent kernel contains entries of order $1\big/\epsilon^2$. Next, they conclude that ODE describing the evolution of residual becomes stiff for sufficiently small $\epsilon$, and this is the main reason why training stagnates. Several numerical experiments support theoretical statements.

**Audience:**

Yes

**Claims And Evidence:**

No

**Requested Changes:**

**Questions:**

1. Claim "We show that if the coefficient in the elliptic operator contains frequencies on the order of $1 \big/ \epsilon$, then the Frobenius norm of the neural tangent kernel matrix associated with the loss function grows as $1 \big/ \epsilon^2$" is not supported.

First, Theorem 3.2 is more restrictive, since it applies only to the periodic functions of the special form $a_{\epsilon}(x):=a(x\big/\epsilon)$.

Second, I am not sure that the claim as stated is accurate. For example, lets consider $a_{\epsilon}(x) = \epsilon \cos(2\pi x \big/ \epsilon)$. For any fixed $\epsilon$ this term contains frequencies of order $1\big/\epsilon$, yet the neural tangent kernel is not going to contain terms of order $1\big/\epsilon^2$ since the absolute value of the derivative is bounded by $2\pi$

I suggest somewhat reformulating the sentence in the abstract or providing a stronger theoretical result.

---

2. Theoretical result is too narrow. It is not clear whether it is applicable in a broader context.

As I understand, the authors want to study PINN for elliptic PDEs with oscillating coefficients. Yet the theoretical result applies only to the specific class of PDEs and specific loss. Here are several suggested directions that can make the article applicable to a broader set of problems:

a. What if $a_{\epsilon}(x)$ is such that the derivative is bounded? Is it going to affect the training or not? For example, we can consider a rescaled version of equation (28) from the article:

$$
-\frac{d}{dx}\left(\frac{\color{red}{\epsilon}}{2.1 + 2\sin(x\big/\epsilon)} \color{black}{\frac{d}{dx} u(x)}\right) = f^{\epsilon}(x) \color{red}{\epsilon}
$$

Authors observed poor training with a version that is not rescaled. After the rescaling tangent kernel does not contain $1\big/\epsilon^2$. Will this simple rescaling improve training?

b. Consider a non-conservative form of elliptic equation

$$
a_{\epsilon}(x)\frac{d^2}{dx^2} u(x) = f(x).
$$
Now tangent kernel again is well defined for $\epsilon\longrightarrow 0$ as long as $a_{\epsilon}(x)$ is uniformly bounded. Is it right, that the training will be efficient for strongly oscillating $a_{\epsilon}$?

c. What if we use the variational form (e.g., as in the Deep Ritz method)?

In this case, we do not need to consider the derivative of $a_{\epsilon}$. Does it mean that the neural tangent kernel is well-defined? Are we going to see poor performance of PINN?

---

3. Experimental evidence is not sufficient.

The approach taken by the authors in Section 4.2 is quite elegant. Yet, the results are not convincing. I suggest repeating the same experiments (learning in $L_2$ setting, PINN+Poisson, PINN+Darcy) for a range of $\epsilon$ and showing errors for different methods.

Besides, I suggest adding at least rescaled version of equation (28) as explained in 2. above.

---

4. Minor suggestions.

a. In equation (6), it is not clear right away that the index $k$ refers to the coefficient with frequency $k$. Please consider clarifying this part.

b. In Figure 1, a logarithmic scale in $y$ seems more appropriate.

c. Neural tangent kernel should be symmetric positive semidefinite. Yet in Lemma 3.1 $K_{ub}\neq K_{bu}^{T}$. It seems there is a typo in the coefficients.

d. Page 6 "Although it does not hold for the ReLU function  $\sigma(x) = \max(0, x)$, these are of course not suitable for NN-based solutions to PDEs with second order (or higher) differential operators." Perhaps, only for a strong form of PDE. Neural Networks with ReLU functions form a nonlinear approximation space of piecewise linear functions with variable positions breaking points. They cover the space of linear finite elements.

---

I encourage the authors to respond to the three main points above with additional experimental data or theoretical results, or rebuttal. In general, I believe that the author's contribution is valuable and suitable for publication in TMLR after the above issues are settled, one way or another.

**Strengths And Weaknesses:**

An article is a pleasure to read. It tells a coherent story, contains good references, and clearly explains the problem. It is possible to improve the presentation in several places, but overall the results are well-explained. In addition to theoretical results, the article contains a few simple experiments that support the claims made by the authors.

On the negative side, obtained theoretical results contain somewhat artificial assumptions. It is unclear to what extent the author's findings explain the pathological behavior of physics-informed neural networks applied to problems with oscillatory coefficients.

Besides, I cannot agree with the central claim from the abstract "We show that if the coefficient in the elliptic operator contains frequencies on the order of $1 \big/ \epsilon$, then the Frobenius norm of the neural tangent kernel matrix associated to the loss function grows as $1 \big/ \epsilon^2$". The results do not seem to support it.

---

> ### Author Response · Authors · 2023-08-31
> **Author response**
>
> We sincerely thank the referee for the thorough review and the myriad suggestions to revise and improve the manuscript.
>
>      --Q1 response--
>
> Thank you for pointing out this oversight in our formulation. Indeed, to make the claim more rigorous, one needs to stipulate that the coefficient $a^{\epsilon}$ is uniformly elliptic in $\epsilon$. In one dimension, this means we require $\forall \epsilon > 0$ that
> $$
> 0 < \lambda \le  a^{\epsilon}(x) \le \Lambda < \infty  \qquad \forall x \in \mathbb{R}
> $$
> where $\lambda$ and $\Lambda$ do not depend on $\epsilon$. Indeed, in the case $a^{\epsilon}(x) = \epsilon \cos(2\pi x/\epsilon)$,
> the elliptic problem described in Eq.~(15) in the paper is not well posed as $\epsilon\to 0$.
>
>      --Q2 response--
>
> (a) This is an excellent question, and we expanded our numerical experiments to characterize the results of rescaling in this way.
>
> The rescaling is equivalent of course to multiplying the PDE residual portion of the PINN loss function by $\epsilon^2$, and it is indeed quite sensible, as it cancels out the $1/\epsilon^2$ scaling the $K_{uu}$ NTK matrix subblock.
>
> In our expanded numerical results section, we observe that the Darcy PINNs without this rescaling simply do not satisfy the correct Dirichlet boundary conditions; the imbalance between the two different terms in the loss function (corresponding to the PDE residual and the boundary conditions) is too great. This imbalance is already an issue for Poisson-type equations, but the problem is exacerbated in the Darcy case by the $1/\epsilon^2$ NTK scaling.
>
> As discussed extensively in Wang et al.\ (J.\ Comput. Phys., 2022), both the magnitude and the distribution of the eigenvalues of the $K_{uu}$ and $K_{bb}$ NTK matrix subblocks are closely connected to a PINNs performance.
>
> Although rescaling the loss function of course changes the magnitude of the eigenvalues of the $K_{uu}$ subblock, indeed it does not change their distribution. As shown in Figure 3, there is a large discrepancy between the principal eigenvalue of the $K_{uu}$ subblock and the others, which is a characteristic sign that the gradient descent dynamics (Eq.~(11)) are stiff. In our new numerical tests of rescaling the loss function by $\epsilon^2$, we still observe poor training behavior. While the Dirichlet boundary conditions are satisfied at a larger $\epsilon$ value (which is *not* the case if the loss function is not rescaled), this does not hold at smaller values of $\epsilon$,
> which we attribute to the NTK matrix spectrum.  Please see the new Section 4.4 for more details.
>
> (b) This is a great point; in non-conservative form, the NTK matrix subblock $K_{\epsilon,uu}$ indeed does not scale as $1/\epsilon$
> as it does in the conservative-form case. In the one-dimensional case, however, because $a_{\epsilon}(x)$ is coercive, we can simply
> recast the equation
> $$
> a_{\epsilon}(x) \frac{d^2}{dx^2} u(x) = f(x)
> $$
> as
> $$
> \frac{d^2}{dx^2} u(x) = f(x)/a_{\epsilon}(x) =: g_{\epsilon}(x)
> $$
> and we are left with a multiscale Poisson-type equation, as considered in the numerical examples and previously analyzed by Wang et al. (see references). In higher dimensions, however, we note this observation is no longer valid.
>
> We modified the abstract and the text to emphasize that the paper develops theory for the conservative form case. Because equations of this form commonly arise due to physical conservation laws, we hope this is of sufficient interest to stand on its own.
>
> (c) This is another excellent point; the focus of the present work is on PINNs, however, in alternative NN based approaches to solving PDEs based on energy-minimization, such as the Deep Ritz method, the derivative of $a_{\epsilon}(x)$ indeed does not appear. Li, Xu, \& Zhang (Commun.\ Comp.\ Phys., 2020) applied the Deep Ritz method in combination with specialized activation functions to multiscale elliptic problems in divergence-form considered here (as well as to nonlinear problems), with mixed results.
>
> To the best of the authors knowledge, a NTK theory has not been developed for the Deep Ritz method, but this is an interesting avenue for future research. We added some discussion along these lines to the Conclusions section of the manuscript.
>
>      --Q3 response--
>
> In retrospect, it was foolish not to include a study for decreasing $\epsilon$ in the original submission. Thank you very much for the suggestion.
>
> We indeed carried out a sequence of additional numerical experiments for the Darcy, Poisson, and $L^2$ regression cases and reported the results in the new Section 4.3. We also added a study as a function of $\epsilon$ of the Darcy PINN rescaled by $\epsilon^2$, as suggested in Comment 2a; please refer to the new Section 4.4 for the results.

---

> > ### Author Response · Authors · 2023-09-04
> > **Author's response, continued**
> >
> > --Q4 response--
> >
> > (a) Clarified. Thanks!
> >
> > (b) Figure 1 has indeed been switched to now have a log-scale on the $y$ axis. The frequency principle is still apparent, but because things are visually crowded, the authors have a slight (but not strong) preference to revert to a non-log scale, however, this is but a minor point of contention.
> >
> > (c) In general, while the $K_{uu}$ and $K_{bb}$ subblocks of the NTK matrix for PINNs are symmetric and positive semi-definite, the $K_{ub}$ and $K_{bu}$ subblocks are not square unless the collocation points in the domain interior $N_c$ and boundary $N_b$ are equal. Because of the scaling of the two different terms in the loss function (equation (8)), the $K_{ub}$ subblock ought to be the scaled transpose of $K_{bu}$.
> >
> > (d) We indeed modified the sentence in the text to specify NN based solution methods (such as PINNs) that are based on the *strong* formulation of PDEs. Thanks for pointing this out.

---

> > > ### Comment · Reviewer_c8m4 · 2023-09-15
> > >
> > > In the revision, the authors resolved most problematic parts of the original submission discussed in my review. In particular, the expanded section with training examples contains the requested experiments, and the corrected abstract appropriately describes theoretical contributions.
> > >
> > > Because of that, I support the publication of the revised manuscript in TMLR.
> > >
> > > Below is the list of questions regarding the new changes. This new list is chiefly on stylistic or minor issues, so the discussion of these additional questions (or the lack thereof) will not change my decision.
> > >
> > > 1. The expanded discussion of experiments is a little hard to follow. I suggest adding a table or figure with average generalization error norms (maybe also a standard deviation) for each method and each $\epsilon$. The particular shapes of the solutions obtained by PiNN do not seem to provide much insight. These plots can, perhaps, appear in the appendix.
> > > 2. I agree that the log scale does not improve Figure 1.
> > > 3. I suggest clarifying (in the vicinity of Lemma 3) that the loss function differs from the one in the referenced article [1], so the full NTK matrix is not symmetric. Surprisingly, in the second referenced article [2], the loss is the same as in the present article. Still, the authors of [2] treat NTK as symmetric and claim that $K_{ub} = K_{bu}$ ($K_{ur} = K_{ru}$ in their terms, equation 3.16).
> > >
> > >
> > > [1] -- https://arxiv.org/abs/2007.14527
> > >
> > > [2] -- https://arxiv.org/abs/2012.10047

---

> > > > ### Author Response · Authors · 2023-10-10
> > > > **Author response**
> > > >
> > > > 1. We added two tables to section 4.3 to summarize the key points from the additional numerical experiments at smaller values of $\epsilon$, and we moved the plots of the PINN errors to the appendix.
> > > >
> > > > 2. We reverted back to the previous, non-log version.
> > > >
> > > > 3. Indeed, immediately after Lemma 3.1 we emphasize that only the $K_{uu}$ and $K_{bb}$ subblocks of the NTK matrix are symmetric, as the scalars $\lambda$, $N_c$ and $N_b$ change the off diagonal blocks. We elected not to explicitly compare to the presentation in article [1] linked above (Wang, Yu, & Perdikaris), since in some places in their work the full NTK *is* symmetric (e.g. in equation (3.4)) but in others, it is not (see, e.g., equation (6.2)).
> > > >
> > > > Thank you again for your helpful suggestions to improve the manuscript.

---

### Decision · Action_Editors · 2023-09-25

**Recommendation:** Accept as is

**Comment:**

Novelty was a concern in the paper, as pointed out by one reviewer, given existing studies that show the difficulty of training imbalanced multi-objective loss. However, the authors clarified that the present work focuses on elliptic problems featuring a more general class of divergence-form differential operators with oscillatory coefficients compared to related studies. This discussion has been added in the revision, making the position of this paper in the literature clear. Moreover, the authors have addressed other concerns raised by the reviewers. According to the evaluation criteria, evidence supporting the claim is more important than novelty and significance. The paper meets the acceptance criteria. Please take into consideration additional comments raised by Reviewer c8m4, which do not affect the decision, when preparing the final version.

**Audience:**

This paper could be of interest to parts of the TMLR audience.

**Claims And Evidence:**

The authors investigate the pitfalls of Physics-informed neural networks (PINN) when solving multiscale elliptic PDEs. They claim that training these networks with gradient-descent-based methods becomes increasingly difficult for highly oscillatory problems. This claim is theoretically supported by demonstrating that the Frobenius norm of the neural tangent kernel matrix grows as $1/\epsilon^2$ when the coefficient in the elliptic operator contains frequencies on the order of $1/\epsilon$. Several numerical experiments support these theoretical statements.

---

> ### Author Response · Authors · 2023-10-10
> **Final edits**
>
> We made a couple of final edits to the manuscript, as suggested by reviewer c8m4 below.
>
> On behalf of the authors, thank you for handling our manuscript.